# Urban Remote Sensing with Spatial Big Data: A Review and Renewed Perspective of Urban Studies in Recent Decades

**Danlin Yu** [1,*] and **Chuanglin Fang** [2]

1 Department of Earth and Environmental Studies, Montclair State University, Montclair, NJ 100101, USA
2 Center for Urban and Regional Planning Design and Research,
Institute of Geographic Science and Natural Resources Research,
Beijing 100045, China
* Correspondence: yud@montclair.edu; Tel.: +1-973-655-4313

**Abstract:** During the past decades, multiple remote sensing data sources, including nighttime light images, high spatial resolution multispectral satellite images, unmanned drone images, and hyperspectral images, among many others, have provided fresh opportunities to examine the dynamics of urban landscapes. In the meantime, the rapid development of telecommunications and mobile technology, alongside the emergence of online search engines and social media platforms with geotagging technology, has fundamentally changed how human activities and the urban landscape are recorded and depicted. The combination of these two types of data sources results in explosive and mind-blowing discoveries in contemporary urban studies, especially for the purposes of sustainable urban planning and development. Urban scholars are now equipped with abundant data to examine many theoretical arguments that often result from limited and indirect observations and less-than-ideal controlled experiments. For the first time, urban scholars can model, simulate, and predict changes in the urban landscape using real-time data to produce the most realistic results, providing invaluable information for urban planners and governments to aim for a sustainable and healthy urban future. This current study reviews the development, current status, and future trajectory of urban studies facilitated by the advancement of remote sensing and spatial big data analytical technologies. The review attempts to serve as a bridge between the growing "big data" and modern urban study communities.

**Keywords:** urban studies; meta-analysis; remote sensing; spatial big data; spatiotemporal data analytical strategies

## 1. Introduction

Urban space is the most important man-made spatial structure on the surface of this planet. The history of urban development signifies the trajectory of human development, technological advancement, the industrial revolution, political system evolution, and even human nature as a gregarious species that enjoys staying together for both security and prosperity.

### 1.1. Data Collected for This Review

Studies of the urban space [1–6], urban landscape [7–21], urban demographic dynamics [22–32], urban infrastructure [33–38], and urban ecological functions [39–48], among many other topics, are abundant and various. A simple search using the keyword "urban studies" (29 November 2022) in the Web of Science® database resulted in 360,368 entries, with many of the results in the fields of "Environmental Sciences" (21.17%), "Environmental Studies" (9.65%), and "Public Environmental Occupational Health" (9.25%). Studies that fall within the categories of "Urban Studies," "Geography," and "Regional Urban Planning" also count for a combined 16.82% of the total search. Figure 1 shows the meta-analysis of this search (with the top 15 categories) conducted on 29 November 2022.

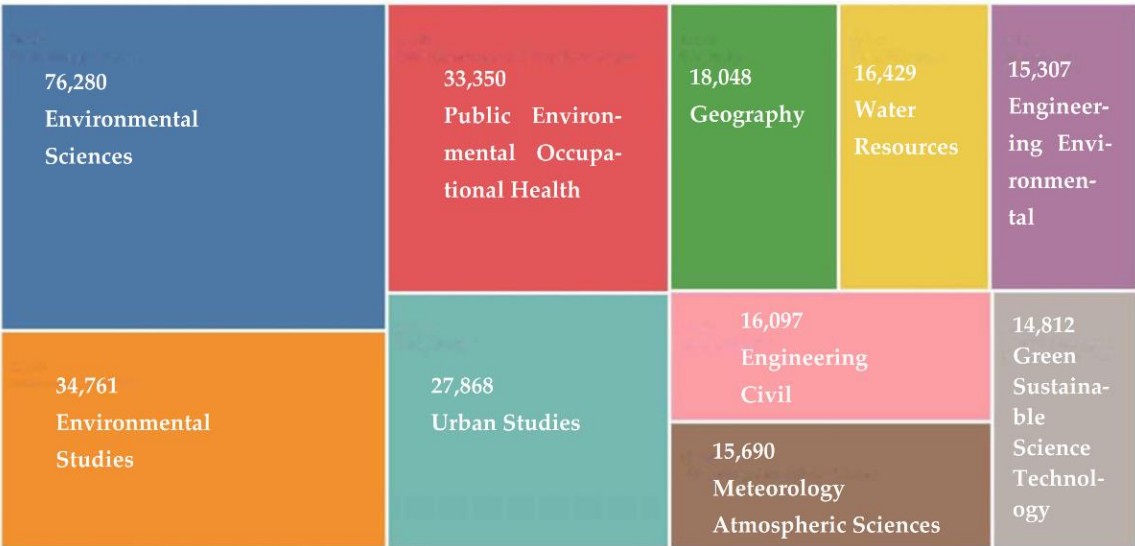

**Figure 1.** Results of the Meta-analysis of "Urban Studies" in the Web of Science Database (29 November 2022).

This broad search result suggests a fundamental core topic and current research foci in the field of urban studies. From Figure 1, the many research topics in urban studies seem to be in the fields of environmental sciences and studies. This might signify a research paradigm shift from an early focus on the cities themselves, transitioning from understanding and describing the varied urban patterns, the vibrant urban landscapes, the burgeoning urban development, and the dynamic urban socioeconomic complex, to more research on the complex and intensive human–nature interaction of the urban space. Many current urban studies, therefore, aim to understand the fundamental mechanisms of the human–nature interaction, the means and tools to reach sustainability, and the practices and policies that can guarantee long-term prosperity [49–54]. This is not surprising, though, as urban space is the place where the most intensive interactions between humans and nature occur. The concentration of large numbers of people on a limited land area quickly causes numerous issues that manifested first as "sick" urban dwellers [55–62], "dirty" cityscapes [55,57,63–67], "crowded" neighborhoods [68–75], "overburdened" urban infrastructure [37,49,76–85], and then eventually as "rapidly degraded" urban environments [55,86–101]. Issues regarding environmental degradation and calls for sustainability in the urban space emerged quickly after the call for "Our Common Future" [102] in 1988. As a response to this, scholarly attention gradually shifts to the environmental and ecological services that cities need to supply in order to provide continuous quality of life for both the urban dwellers and the nature where the urban spaces are located.

Research foci are of critical concern regarding the development directions of a particular subject area (urban studies here). In addition, successful studies also rely on reliable, dynamic, high quality data sources, and robust, effective, and scrutinizing methods that can facilitate not only the understanding of a particular case but also the interpretation of the general theoretical framework of urban studies. Of the contemporary urban studies research themes, two of the most popular dynamic, reliable, and high-quality data sources are the objectively collected remote sensing images, and spatial big data generated by geotagged search engines and social medial platforms. A rising research method in urban studies is the spatial and spatiotemporal data analytical methods.

Applying the rapidly accumulating remote sensing images (including satellite, aerophoto, and on-demand drone images) and spatial big data in urban studies is critical to advance urban theories and support urban sustainability research themes in the digital era [103–105]. Urban spaces are highly dynamic and highly functionally organized because of the dense concentration of people and human activities. Traditional studies of cities rely heavily on

field surveys, observations, and yearly censuses. It was, therefore, critical to explore cities from a system perspective [1,103,106,107]. While the systematic view of cities is important for urban planning and long-term development, investigating cities from a more detailed and higher resolution (in both spatial and temporal dimensions) perspective with advanced computational algorithms provides a refreshing theoretical understanding of everyday urbanism [108–110]. The development of urban space is a critical land use–land cover pattern change from agricultural and low residential density land use to an impervious and highly compact land use pattern. Assisted by the availability of public remote sensing data, such as Landsat images, and advanced machine learning algorithms, studies are able to monitor urban expansion dynamics [111], assess the changing trajectory of rural town settlements' land use [112], and monitor urban land use changes to support smart urban growth [113]. The recent booming of urban vibrancy studies with satellite images and social sensing spatial big data suggests the urban study community is quickly catching up with the new data and the new approaches [114–122]. This is a manifestation of everyday urbanism in action in the remote sensing and spatial big data era for urban science.

### 1.2. Scholarly Attention on Remote Sensing in Urban Studies

Remote sensing technologies have experienced unprecedented development over the past decades, thanks primarily to sensor advancements and continuously increasing information infrastructure [123]. One of the key advancements in remote sensing technology development, and closely related to urban science, is object detection from remote sensing images. After an intensive review of recent deep learning-based object detection progress, Li, et al. [124] proposed a large-scale, publicly available benchmark for object detection in optical remote (DIOR) sensing images, which contains 23,463 images and 192,472 instances, covering 20 object classes. The benchmark established the baseline for scholars to develop and validate their own study, which is particularly useful in urban science. Furthermore, scholars have reviewed remote sensing applications in urban studies specifically from a variety of aspects. Urban remote sensing is most commonly applied and reviewed in urban ecosystem services, urban environmental monitoring, urban land use mapping, green spaces, and urban heat islands [125–129]. Neyns and Canters [130] reviewed the development of mapping urban vegetation using remote sensing techniques. Wang, et al. [131] reviewed studies using remote sensing and GIS technologies on urbanization and urban development. Evaluation of spatial big data in urban studies seems to be just about to emerge. Hao, et al. [132] reviewed the application of spatial big data for urban planning practices in China. Yin, Dong, Hamm, Li, Wang, Xing and Fu [104] reviewed studies that combine both remote sensing and spatial big data in urban landscape mapping. These reviews provide insightful comments on the development and integration of remote sensing and spatial big data in urban studies from various aspects, including urban ecoenvironment, urban landscape mapping, urban planning, urbanization, and development, among others. A comprehensive review of how remote sensing and spatial big data, as well as spatial–spatiotemporal research methodology, influence the research of urban science as a whole, however, merits further investigation. This current review aims to fill in this gap. Figure 2 charts the structure of this review, with a focus on advancing urban science with remote sensing images, spatial big data, and spatial–spatiotemporal analytical methods. A further database search with the keywords "remote sensing," "big data," and "spatial analysis" in the Web of Science® suggests that only a handful of studies (167 entries, 29 November 2022) contain all three keywords in "any fields."

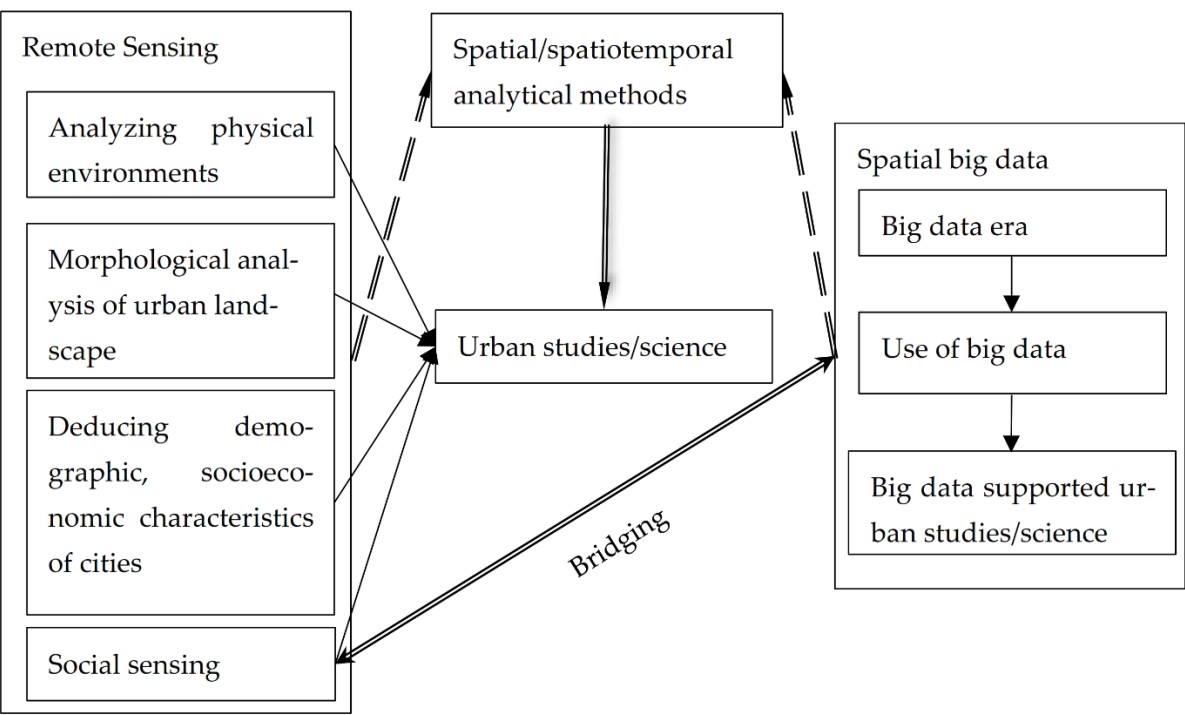

**Figure 2.** Flowcharts of remote sensing, spatial big data, and analytical methods supported urban studies/science.

Clearly, while research in urban studies now primarily falls within the fields of environmental sciences and studies, and focuses mostly on sustainable development, agenda, approaches, action plans, and strategic operations, works that take advantage of the most recent developments in observational technology (remote sensing), geotagged data generating platforms (spatial big data), and advanced spatiotemporal data analysis techniques (such as spatial econometrics and Bayesian hierarchical spatiotemporal modeling, among many others) are only starting to take off. This review focuses on urban studies that closely integrate with varied remote sensing technologies, including sensors on satellite observatory platforms, aircraft-carried sensors, and unmanned aircraft systems, coupled with the newly developed "remote sensing" data acquisition technology. These include the spatial big data generated in the new era with rapidly growing geotagged social media postings, search engine queries, and content analysis from online documents and reports. We will also review some of the newly developed spatial and spatiotemporal data analytical frameworks for urban studies, especially the Bayesian hierarchical spatiotemporal models that can provide robust modeling results and take full advantage of the potentials of the traditional geographic knowledge (through the definition of informative priors) and newly emerged remote sensing information and spatial big data. Our goal is to provide a renewed viewpoint of urban science in the era of booming remote sensing technology and spatial big data. It is, however, not the authors' intention nor our capacity to provide an exhaustive review of all possible viewpoints and methodological advancements in the recent development of remote sensing, spatial big data, and their applications to urban studies. For instance, studies that cover the most recent advancements in machine learning algorithms in urban science with remote sensing and spatial big data can be found in Li and Dong [133], De Nadai, et al. [134], and Ma, et al. [135], among others. This topic will not be covered in this report.

This review is organized as follows. After this introduction section, we attempt to provide a brief review of the definition of urban space and urban studies prior to the emergence of remote sensing technologies. This is followed by a review of the varied urban remote sensing studies up until this day. We then define spatial big data sources as newly

emerged "remote sensing" data source that does not rely on a specific electromagnetic spectrum signature but on the widespread social media postings and web searches that are geotagged. Integrating remote sensing and social sensing data in an urban science context is also discussed. This is followed by a detailed introduction to the Bayesian hierarchical spatiotemporal modeling framework and how it will be utilized in the integration of remote sensing technologies and urban studies for a sustainable urban future. The manuscript concludes with a summary of the review, a meta-analysis, and an outlook for better urban remote sensing studies in the future.

## 2. Gregarious People, Compact Lands, and the Development of Cities

Harsh prehistoric environments taught human beings that an organized, larger group had much better chances to survive than individuals or smaller groups [136]. One of the cornerstones of human development was the realization that a division of labor among a group of inhabitants in a place could enhance the group's productivity beyond the simple sum of the individuals' outputs [137]. In their research of the Roman Empire, Hanson and colleagues [138] create a model to establish a link between the division of labor and the urban space that demonstrates its social networks in its built environments. Scholars identified that a higher level of functional diversity of human settlement in a place is key to a better survival strategy for the settlement [137–139]. The highest levels of functional diversity are often observed in sites of ancient cities, and many still exist until this day, such as Rome of the Roman Empire, Beijing and Nanjing in China, Cairo in Egypt, and Athens in Greece, to name but a few.

Functional diversity, therefore, is the primary characteristic of any cityscape and an early focus of urban studies.

Early studies of such integrated functional diversity in cities started with theories regarding the spatial organization and the division of labor. In *The Isolated State* [140], von Thünen observed how locations of human habitats were organized as a circular spatial structure (which he termed the "Thünen Circle").

This theoretical model of the spatial structure of any human habitat was later adopted by urban geographers Christaller, Lösh [141–144], and many others, to develop the theory the of central place of cities. The central place theory suggests that urban spaces are hierarchical in nature with the central place having the most comprehensive functions, while peripheral locations have more specific functions when moving away from the center. For instance, via estimating the microfoundations (the threshold and range of good) of the central place theory through data obtained from social media platforms Foursquare and Twitter in Louisville, Kentucky, van Meeteren and Poorthuis [145] found that urban central functions have typical ranges and thresholds related to population spread, and that central functions has an approximate hierarchical structure. Similar findings were reported in [146] while using spatial big data to identify local and nonlocal functions in the urban centers of Shanghai. It was found that urban centers have a strong presence of both local and nonlocal functions, while the peripheral regions tend to have more local functions. During this period of theoretical development, studies on cities focused primarily on the urban spatial structure, which was primarily observed and explored through fieldwork, analysis of the spatial structures of existing cities, and mathematical models that originated from the "Thünen Circle."

Due to limited computational power and observational data sources at the time, the spatial structure of cities was often assumed to follow a near perfect geometric shape (under ideal conditions), describing the spatial pattern and seeking the mechanisms behind such a geometry became dominant research themes during the early stages of urban studies [141,142,144]. Infrastructural constructions such as telephone connections, road connections, transportation costs, terrain, closeness to water, land, and mineral resources, and even temperature and precipitation, were all considered as urban development mechanisms in the long history of urban studies. Christaller [144] used "telephone connections" to measure the "centrality" of a place, hence defining the hierarchy of an urban system in

the early 20th century. Later, Hannerberg, et al. [147] utilized shop sizes, and Taaffe [148] relied on flights to define urban centrality and hierarchy. Many of these models relied on simple geometric explorations under ideal conditions (such as flat terrain, homogeneous transportation costs, and free movement of production factors).

Rapid urbanization worldwide after the second world war, however, had rendered both the models and existing observational practices (primarily through survey and statistical census) less adequate to discuss and study the urban landscape and its dynamics. While urban sprawl [149,150] started to attract attention due to the widespread availability of automobiles [150], and sustainability studies became a central theme of scientific investigations right after the Brundtland's *Our Common Future* [102], it became clear that new strategies, new methods, new research paradigms, new observational means, and new models are needed to understand the ever-growing, highly dynamic, and intensely complex cities.

### 3. Remote Sensing and the Advancement of Urban Science

In the early development stages of remote sensing technology, the term "big data" was not on the horizon. Back then, applications of remote sensing technology were primarily for observation, change detection, and information extraction, limited by the available spatial and temporal resolutions [151]. The rapid development of various sensors and the accumulation of remote sensing images in the recent decade, coupled with the recognition of us entering a "big data" era, however, has greatly changed the ways remote sensing images are stored, processed, analyzed, and utilized. In their study, Xu, et al. [152] regard remote sensing as a form of "big data" (remote sensing big data) and proposed a modular framework attempting to connect the data (remote sensing images) and computation (big data computation). This is especially effective with the advancement in computer science and computational capabilities of today's networked hardware and software environment. Consequently, Xu, et al. [153] argue that cloud computing is an effective way to activate and mine large-scale heterogeneous data such as remote sensing big data. In addition, Zhang, et al. [154] study suggests that deep learning algorithms are effective and efficient ways to process and analyze remote sensing big data, including geometric and radiometric rectification and processing, cloud detection and removal, data fusion, object identification and extraction, land-use and cover classification, change evaluation, and multitemporal analysis. The coupling of remote sensing and big data starts off with a mutually supportive relationship. While accumulative remote sensing images are undoubtedly a form of spatial big data, spatial big data also extends its horizon to include data acquired from geotagged social sensing, in which the sensors are none other than the people who are also part of the dynamic urban space complex.

This is when remote sensing and spatial big data and their relevant processing and analytical frameworks become indispensable for advancing urban science in the immediate future [152,155,156]. We detail the development trajectories of both remote sensing and spatial big data and how they gradually embed in contemporary urban science/studies in the following sections.

### 3.1. Remote Sensing and Its Application to Urban Studies

The term, "urban remote sensing," or, rather, applying remote sensing technologies to study urban phenomena and urban environments, only appears in the late 1950s. Norman and colleagues started to explore the urban environments in the late 1950s using aerophotos to interpret the social structure, human geography, and human ecology of cities [157–159]. A report submitted to NASA and the Geological Survey [160] attempted to use color infrared aerial photos to analyze urban residential environments in the Los Angeles basin. As meticulously noted in their report, the authors stated that applications of remote sensing techniques in urban studies were slower than in other fields such as land use land cover change detection, water resource management, and forest management. They argued that this was because of the "great diversity of the urban environment," and the "complex

nature of the spatial relationships" among different urban elements. In addition, the remote sensing techniques at the time were also limited by the available spatial and temporal resolutions of the remote sensing products that were typically coarse for typical urban applications. Urban environments, unlike in other fields where remote sensing found lively applications, require much smaller spatial and much shorter temporal resolutions to produce meaningful and actionable study results.

Still, the sheer volume of information that is contained within the remote sensing products (even though most of such products are in physical paper formats, and often produced by airplane-borne unstable sensors for urban application), was very tempting for urban scholars, especially since such approaches provided timely and abundant information that traditional approaches fall short on, such as large area land use change detection [8,161–167], urban waterbody and green space extraction and mapping [168–173], urban environmental justice evaluation [171,174,175], and urban heat island detection and mechanism studies [176–188], among many others. Based on our meta-analysis that combs through the last half century's studies, the application of remote sensing technology in urban studies could be roughly divided into four particular aspects that we will detail below.

### 3.1.1. Extracting and Analyzing Physical Environments of Urban Areas

Using remote sensing images (be it aerophotos or satellite borne sensed images) to detect land use land cover change, detection of environmental condition changes, and monitor urban heat island phenomenon were among the most obvious choices due to the different reflectivity in both panchromatic and multispectral bands of different land use land cover types, and the thermal signatures under different temperatures. A simple search in Web of Science® using "urban remote sensing" and "land use" (in all fields) produced 9474 entries (19 December 2022). Alternating "land use" with "heat island" and "green space" produced 2027 entries and 928 entries, respectively (19 December 2022). These are also the fields in urban studies that do not usually require very high spatial and temporal resolutions. The wide availability of images from landsat multispectral scanner (MSS), thematic mapper (TM), enhanced thematic mapper plus (ETM+), and the recent landsat 8 and 9's operational land imager (OLI) and thermal infrared sensor (TIRS) provides a relatively precise 30 m spatial resolution and 16-day temporal resolution. Analysis of these landsat images is often sufficient to produce meaningful results [189,190].

Common algorithms that classify land use land cover [168,191–201], detect and monitor the general urban environments [202–210], air quality assessment and pollution hotspot identification [211–214], and extract the percentage of impervious surfaces [181,215–227] are widely applied in this aspect of urban studies. This is understandable since the practices are a natural extension of applying remote sensing techniques to study natural environments. However, urban areas are more fragmented, more complex, and fluctuate more often and more irregularly than natural environments. Still, the newly developed machine learning algorithms, such as random forest [228,229], support vector machine [204,230], neural network [163,231–234], deep learning [154,235–237], and estimation techniques, including categorized and regression tree (CART) [238], geographically weighted regression [81,219,220,239], and Bayesian learning [240], among many others, provide an ever increasing arsenal for urban scholars to take advantage of the growing remote sensing datasets, be it regular 30 m spatial resolution multispectral images or sub-meter spatial resolution hyperspectral images. Undoubtedly, applying remote sensing techniques to study urban environments, air quality assessment, and urban land use land covers will continue to dominate the frontline of urban remote sensing scholarly activities.

### 3.1.2. Morphological Analysis of Urban Landscapes

Analyzing urban morphology and detecting urban spatial patterns from remote sensing data is straightforward, and of particular importance for urbanization assessments. As noted in the studies by Zhu, et al. [241], an accurate account of urban morphological

features is "at the core of many international endeavors to address issues of urbanization, such as the United Nations' call for Sustainable Cities and Communities" [241]. The authors developed an open access global urban morphology database and hope to facilitate relevant stakeholders such as the United Nations, scholars whose fields focus on urban morphological changes and sustainability, and governments of many developing counties that do not have the financial, technological, and human capital resources to create a national database to improve their spatial assessments of urbanization such as many in the Sub-Saharan African region [126].

From the late 1980s onwards, urbanization has picked up its pace, especially in developing countries, due to increased globalization and industrialization worldwide. One of the major issues of rapid urbanization, as manifested in the developed world right after the Second World War, is the rapid and uncontrollable urban sprawl that caused the urban centers to decline and suburban and exurban areas to emerge with spider-web-like highway networks. Not only did the decline of urban centers exacerbate the deterioration of urban environments and socioeconomic prosperity in the urban centers and the entirety of urban areas as a whole, but also the natural environments that used to surround the cities fragmented. Natural habitats for many species, including endangered ones, were disrupted, and pristine forests, wetlands, and waterbodies were infringed upon and polluted [12,149,150,242–246]. Morphological analysis appears to be a powerful tool enabling urban scholars and practitioners to understand, monitor, model, and predict the extent of urban sprawl and the change of urban spatial structures [12,81,247,248]. For instance, in [12,248], the authors used different years of landsat thematic mapper (TM) imagery to detect the urban landscape pattern changes in Daqing City, China, an oil-rich resource city, and monitor the morphological impact of urban sprawl (growth of the city proper) on local environments and urbanization process. In another research, Huang, et al. [249] conducted a systematic comparison of urban forms between developed and developing countries using satellite images of 77 metropolitan areas in Asia, the US, Europe, Latin America, and Australia to calculate seven spatial metrics that capture five distinct dimensions of urban form. Their results suggest that there are clear morphological differences in urban forms in developed and developing countries, and a sustainable urban form could look different under different socioeconomic development levels. This result is critical to support the key ideas of sustainable urban development in that urban sustainability is a term that is only meaningful when the principle is situated within local conditions [54,250].

Recent adaptations of drone technology in studying microurban forms start to emerge. While drone applications are often seen as a supplementary strategy to monitor the atmosphere [251], the water environment [252], and urban land surface temperature [253], in the study by Loggia and Govender [254], the authors combined drone mapping and field survey to conduct urban morphology investigations and urban mapping in a hope to understand how urban spaces can be effectively captured and interpreted. Urban morphological studies will likely continue to rely more on satellite-based high-resolution images. Other than the commonly available, fixed-schedule satellite high-resolution images, we envision the images produced by a more schedule flexible, sensors on demand, and cost-effective drone fly missions will play increasingly important, and even central, roles in future urban morphological, urban environmental, and urban land cover and land use studies.

### 3.1.3. Deducing Demographic, Social and Economic Characteristics of Cities

While a healthy urban environment and accurate account of urban morphology are surely critical for a sustainable urban future, the urban complex is a myriad of intermingling environmental, social, demographic, economic, and physical build-up, and unique land cover (impervious surface) elements, among many others. At the center of the urban environment are the urban residents and all of the activities that are caused by or occurring around them. A sustainable urban future only makes sense when there is a harmonic relationship and a virtuous relationship between the urban dwellers and the

urban environment. Scholars, especially those in the social science and humanity fields of studies, also attempted to utilize remote sensing techniques in their respective domains. For instance, using remote sensing techniques to estimate the population in an urban area was an early attempt to capitalize on remote sensing images' convenient accessibility and cost-effectiveness compared to a full-scale census or even a 1% or 5% demographic survey (such as the American Community Survey conducted annually). Wu and Murray [255] attempted zonal and pixel-based models with landsat enhanced thematic mapper images to generate population estimates in Dayton, Ohio, and found the estimated population agrees with the common census reasonably well. Similar strategies were applied to estimate the residential population in China at various scales using the impervious surface and dasymetric mapping methods, though accuracy fluctuates at different scales with county level data presenting the best results [256]. While it might sound impractical or even hardly possible that counts of people or other societal, economic, or human behavioral activities can be obtained from remote sensing images, many of the social science studies that rely on remote sensing techniques often go above and beyond just extracting the obvious reflectivity or thermal information. For instance, Yu, Wei and Wu [219] attempted to use extracted impervious surface and barren land information from a landsat thermal mapper image to account for housing price fluctuation in the city of Milwaukee, WI. They found better environmental quality often increases the estimated prices of individual houses, though such effects vary from place to place. In [69,71], the authors combined remote sensing with social sensing and GIS techniques to the evaluate local quality of life, clearly suggesting that remote sensing data provides an effective way to assess everyday urbanism. This is possible since human activities have exerted significant imprints on the urban elements' electromagnetic spectrum that can be well extracted.

In recent years, other than the common multispectral remote sensing images, nighttime light images collected from the US Defense Meteorological Satellite Program's Operational Linescan System (OLS) sensor (from 1971–2011) and the later NASA launched Suomi National Polar-orbiting Partnership (NPP) satellite and NOAA-20 satellite (since 2018), which carried the visible infrared imaging radiometer suite (VIIRS) instrument and produced day/night band (DNB) data, are attracting much attention in socioeconomic, demographic, and building environmental fields of study. This is due to the fact that the intensity of various modern human activities in a place is closely related to the amount of energy consumed there. Nighttime light emission provides an immediate proxy for the intensity of energy consumption, hence a good proxy for a wide variety of human socioeconomic activities [257–260]. In addition, the new generation of nighttime light satellites with a much finer spatial resolution (130 m), like the luojia1-01, are also providing much needed data that might be more suitable for urban studies [257,261]. While nighttime light remote sensing data have been available since the early 1970s, early studies often focused on using nighttime light remote sensing data as a proxy to map the city [262,263] due to the relatively coarse resolution (2.7 km in spatial resolution) and poor, inconsistent radiometric quality due to the lack of on-board calibration. The improved spatial resolution (375 and 750 m depending on the band, and 130 m for the luojia1-01) and onboard radiometric calibration for the VIIRS instrument greatly enhanced the application scope of nighttime light images in urban studies. It was soon found that nighttime light data was a very promising data source in urban studies to estimate population size [261,264–267], explore the urban socioeconomic landscape [45,268], estimate poverty [257], model urban morphology, expansion, and growth [105,269–272], and investigate urban energy exchange with the environment [188,270,273–276], among many other things. This booming application of nighttime light data in urban studies is understandable. While it is true that there are many sources of illumination during nighttime, most notably moonlight and surface albedo, the light produced from various anthropogenic activities is the most obvious and consistent information. The intensity and density of light distribution are directly related to the intensity and density of human activities. For instance, Chen and Nordhaus [277] examined the usefulness of the VIIRS data in the estimation of economic activity with

both US states and metropolitan statistical areas (MSAs). Not surprisingly, with enhanced spatial resolution and wider coverage, their results suggested that high-resolution VIIRS light data provides a better prediction for an MSA's GDP than for state GDP. This suggests that lights may be more closely related to urban sectors than rural sectors, hence better suited for urban-related studies. A similar study [278] examined the county-level GDP for the US using VIIRS version two data. Also, producing comparable results that nighttime light data could be used as a well-justified proxy for regional economic performance.

Other than economic estimation, the intensity and density of lights are also a good proxy for material stocks of buildings to evaluate the building environment in cities [279]. Peled and Fishman [280] used nighttime light data to delineate distinct built-up areas. In their study, the radiance values of the areas under study provide a good estimate of the built-up volume, compared to more traditional measures. Yet, the nighttime light data is considerably less expensive to acquire, covers a large area, and can be easily benchmarked. A similar logic was also adopted in Zhang, et al. [281] study of urban vibrancy. By proxying urban vibrancy with nighttime light vitality, they were able to establish relationships between urban vibrancy and urban diversity, which seems to be a primary driving force for vibrant and sustainable cityscapes. Clearly, the spatiotemporal availability, constant monitoring of the nighttime light data, and the close-knit relationship between nighttime light and anthropogenic activities [259,282–284] offer promising potential for a more indepth investigation into the dynamics and evolution of the urban landscape.

*3.2. Social Sensing—A New Frontier of Remote Sensing and Interface with "Big Data" Semantics*

The term "social sensing" refers primarily to people's ability to perceive and make inferences about what others think and do in their own environments [285]. In their edited seminal book, *Social Sensing*, Wang, et al. [286] define social sensing to be a set of sensing and data collection paradigms where data are collected from humans or devices on their behalf. In this definition, society as a whole (humans, or devices on their behalf) is the context and object for sensing and sensing the means of data acquirement. This is viewed as a direct result of the proliferation of social media and social network platforms such as Facebook, Twitter, LinkedIn, Sina Weibo, Google Search, and Baidu Search, among others. The recent outbreak of the COVID-19 disease has further accelerated the use of these social media platforms to facilitate data acquisition and database construction, which in turn provides powerful means to fight back against the spread of the disease [287,288]. Aggarwal and Abdelzaher [289] presented a broad overview of social sensing and suggested that the growing availability of such socially sensed data provide a natural way to predict and monitor individual as well as societal behaviors, trends, and patterns. The rise of social sensing, coupled with the embedded geotag capabilities via embedded GPS of ever-increasingly available smart devices, and the internet-enabled data sharing mechanism, enabled the arrival of a context-aware computing environment, which proves to be particularly useful and relevant in urban studies [290].

In the seminal collection of dynamic social network modeling and analysis work in 2002 [291], the national research council brought together scientists from social network analysis research to provide sound models and applications with social network generated data. The social network theories, dynamic social networks, metrics and models, and networked world are the four themes that many practitioners from computer science, database development, and social media platforms have devoted their time to discussing. The emergence of social networking and advanced computational capacity, including newly developed machine learning algorithms, were the fundamental drives for the increasing capability to mine indepth rhythms, patterns, and vibrancy of human activities in any type of significant human settlements, cities in particular. Mining the seemingly chaotic and confusingly large amount of social network and social media data became not only possible but critical for decision making. For instance, in 2009, Google Inc.'s engineers and scientists in the Center for Disease Control and Prevention mined google search entries to accurately predict the spread of influenza one or two weeks earlier than the CDC report [292]. In

another work, Eagle and Pentland [293] introduced to the 6th International Symposium on Mobile Human-Computer Interaction a mobile system for sensing complex social systems. They demonstrated the system by collecting data from 100 mobile phones over the course of nine months. Using the collected data, they were able to measure information access and use in different contexts. These measurements allowed them to recognize complex social patterns in a daily manner, infer relationships among individuals and their surroundings, identify socially significant locations, and model organizational rhythms. The potential of geotagged mobile data is boundless, especially in the social science domain, though the study only follows 100 mobile phones over nine months. Similarly, Adams, et al. [294] proposed an online algorithm to infer social context through extracting information from GPS traces. Social rhythms, patterns in time, duration, place, and human beings' reactions to real-world activities and physical environments can all be extracted with amazing accuracy thanks to the more than 10 million samples over 45 man months. Understandably, the human samples in these early studies are limited due to limited computing powers. The algorithms and the context-aware computing framework, however, are easily scalable to larger urban settings.

It remains debatable whether social sensing is a type of remote sensing since remote sensing has traditionally referred to information acquired from electromagnetic energy sensors that collect information generated by electromagnetic energy. Social sensing, however, relies more on individual perceptions and observations of their environments and is facilitated by the rapidly developed telecommunication technology and widely available personal mobile devices with geotagged social medial platforms. In their research, Liu, et al. [295] regarded each individual who supplied information via social media platforms as playing a "role of a sensor," which might be analogous to the electromagnetic energy sensors as in traditional remote sensing. This analogy bridges social sensing with remote sensing, if not regarding social sensing as a form of remote sensing. In addition, they also argued that social sensing information captures socioeconomic features well, while traditional remote sensing information might need complex algorithms and conversions (such as using nighttime light data, high-resolution images for impervious surfaces identification, etc.) to do so [295].

In this regard, we contend that studies of social sensing could be viewed as an extension of studies in traditional remote sensing and that there is a strong need to take advantage of remote sensing analytics and information richness to study more complex and dynamic socioeconomic landscapes, as primarily observed in the urban settings. On the other hand, the arrival and rapid proliferation of geotagged social media platforms and devices that facilitated social sensing allow for a bridge between remote sensing and the next big buzzword, "Big Data" in the new era, and for more rigorous and data-rich urban studies.

### 3.3. Limitations and Challenges of Remote Sensing in Urban Science

While applauding the integration of remote sensing data sources as a great jump in urban studies/science, it is also acutely recognized in especially the urban scientific scholarly community that there exist significant challenges in this new frontier. As pointed out in the early studies by Mullens Jr and Senger [160], spatial resolution is a big hurdle in applying remote sensing technologies to urban science. The spatial resolution of remote sensing data determines the level of detail that can be obtained from an image. For example, satellite images with a low spatial resolution may not be able to capture small-scale features, such as individual buildings or small patches of vegetation. This limitation can be particularly challenging when studying urban areas, where high spatial resolution is often needed to accurately capture the complex and heterogeneous urban environment. Admittedly, more recent remote sensing sensors and equipment, including satellites and unmanned drones, are able to provide sufficient spatial resolution for urban areas. However, the conundrum of cost, availability, and added noise with finer spatial resolution could

quickly amount to a grave challenge for urban scholars to effectively take advantage of this new data source [125,296].

In addition to the spatial resolution conundrum, the temporal resolution of remote sensing images, especially satellite remote sensing, poses an even bigger challenge for contemporary urban science. The temporal resolution of remote sensing data refers to the frequency at which new data is collected. For example, satellite images may only be available once every few weeks or months due to their orbital design, which can make it difficult to capture rapidly changing urban environments, which is particularly crucial for understanding today's urban dynamics [114–117]. This limitation can be particularly challenging when studying urban areas undergoing rapid development or facing environmental issues, or when the study focuses on short time periods where frequent updates are necessary to accurately capture changes over time.

Other than these two major limitations, atmospheric conditions over major urban areas, the spectral resolution of the commonly used multispectral images, and the cost of acquiring high-resolution (spatial, temporal, and spectral) images might prevent urban scholars from fully engaging in integrating remote sensing technologies to urban science. While the advancement of computational capacity, advanced data processing capabilities, and increased availability of remote sensing data certainly show the increasingly important roles remote sensing images play in urban science in recent decades, overcoming these challenges and limitations is still an ongoing process for urban scholars.

## 4. The Emergence of Big Data Thinking, and How Big Data Supports Urban Studies/Science

### 4.1. The Big Data Era

While applying remote sensing information in urban studies has proven to be a long road to trek, the recent buzzword, "Big Data," seems to be naturally suited to studying urban phenomena from the onset. The essence of big data is not necessarily a new concept, though the term was initially used in the early 1990s. From a broad perspective, big data is only relative to the analytical approaches and means (hardware)—collectively the computational capability. When our computing power was low, a dataset that could not be adequately analyzed by the then computational capability was legitimately considered "big data" in the sense that it was too "big" to be processed.

In the precomputer and premodern transportation and telecommunication era, data accumulation and analytical power often went hand in hand in a parallel fashion. While we understood data could be potentially big, the data that concerned us often was within an analytically manageable level. Alternatively, statistical approaches that "sampled" the population satisfied our need to explore and understand the story behind the data. Such an analytical paradigm changed dramatically during the globalization and high-speed, high-powered computational era when clustered computation became increasingly popular for data management and analysis [297,298]. Accumulation of information was explosive, and, while the computational power and analytical power were also growing, it was in no way parallel to the increased amount of information. As a matter of fact, the renowned urban geographer, Batty [103] cited an anonymous source defining "big data" being "any data that cannot fit into an Excel spreadsheet." This is particularly true in urban science since the highly dynamic everyday urban events are now able to be recorded, layered, assessed, analyzed, and incorporated into real-time decision making for a more livable and sustainable urban environment [30,299]. The development of the general idea of "big data" also originated from constantly arising urban development and planning problems that could not be adequately handled by conventional means [300,301], as noted in the seminal book by Mayer-Schönberger and Cukier [302].

From a retrospective point of view, the emergence of "big data" was likely inevitable. Our understanding of the world has always been within our cognitive boundaries. Even with intensive training, unfortunately, such boundaries never extend much. Our means of accumulating data, on the other hand, often are revolutionized once every few decades

to years. During the precomputer era, "what we see and hear are what we get" stood as the standard way of information acquisition. Information dissemination means were also limited to printed paper and often confined within bordered geographies. Our cognitive boundaries agreed well with our accumulated information. "Big data" was there, but never emerged to the surface to become a "big" concern for scholarly minds.

The arrival of computers started to change how information was both accumulated and processed. While in the early stages of computer development, information accumulation, and processing were still somehow developing in a parallel fashion. The development of networked computers in the 1960s, and later the transfer control protocol/internetwork protocol (TCP/IP) which supported the internet and the world wide web in the early 1980s (some argue January 1st, 1983, is the official "birthday" of the Internet), shattered the parallel relationships between information accumulation and processing. Information accumulated in an explosive way as computers across continents were linked together through the "Internet". The arrival of handheld smart devices further increased the gap between information accumulation and information processing capabilities. At this point, we are entering a "big data" era.

*4.2. Big Data Thinking*

It is generally agreed that there are roughly three phases of the concept and understanding of "big data" [303], based on how data is accumulated, stored, and analyzed. The first phase concerns primarily the structured content of information, roughly covering the period from 1970–2000. It was directly linked to the long-standing domain of database management. During this phase, data storage, extraction, and optimization techniques were the foci. In other words, this was when "big data" emerged and the primary task was to understand the fact that "data is coming" and we need to find efficient ways to store it. The prominent development in this phase was the transition from flat-file data storage to hierarchical data storage, to the development of relational database management systems (RDBMS) which is still used today as a standard data storage format to facilitate fundamental data analytics. Data warehousing, data mining through traditional statistical analysis, and dynamic near-real time information updating via online dashboards and scorecards were the primary activities in this phase of big data development.

The second phase of big data development started from the early 2000s to around 2010 when the internet and relevant web applications produced enormous amounts of data. In addition, search engines including Yahoo®, Google®, and Baidu ®, among others, had also produced enormous amounts of web-based unstructured content. Big data development in this phase, therefore, is concerned with primarily exacting regularities from the seemingly irregular, unstructured data. For instance, many big internet commerce companies, such as Amazon®, eBay®, and major online news agencies often analyzed customer behaviors through their click rate, content-viewing trends, search logs, and even IP-address associated geographic locations to generate highly targeted, specific content and recommendations for their customers. The massive increase of data resulting from fast-growing web traffic and the wide reach of the internet globally during this phase demanded more advanced data analytical techniques. Coupled with increased computational power, new network analysis, web mining, and spatiotemporal analysis methods emerged rapidly during this phase.

The third phase of big data development was from 2010 until now. This is the phase when mobile devices (mobile phones, tablets, and mobile workstations, among many others) dominated the consumer electronics market. In 2020, it was estimated that there were 10 billion devices that were connected to the internet [304]. The emergence of social media and mobile browsing and mobile devices' constant connection to the internet, coupled with the embedded GPS tracking device, enables us to collect enormous amounts of data regarding individual behaviors, and movements, and even deduce individual health status, shopping preferences, and detailed daily activity patterns. Not only are the numbers of mobile devices increasing, sensor-based and internet-enabled devices, such as smart TVs,

internet-enabled thermostats, smartwatches, and household appliances, all belonging to this so-called "Internet of Things" (IoT), are also increasing in numbers rapidly. These devices generate huge amounts of data almost constantly as well.

In the face of this volume of massive amounts of data, traditional data analytic approaches often fall short of revealing the rich stories behind the data. In their 2014 seminal book, Mayer-Schönberger and Cukier [302] pointed this out and attributed the difficulty to the fact that traditional analytics primarily focused on finding a causal relationship between different events. Causality might be traceable when our data were sampled from the gigantic but unknown background population data. The traditional means-developed inferential approaches under reasonable assumptions which could be used for causality seeking. While in the face of the current "big data" era, the gigantic background population data is now all of a sudden present in front of us. Seeking causality becomes insurmountably difficult. Instead, it might be more reasonable and practical to seek correlation rather than causality [106,302,305]. Still, despite the difficulty of seeking causality among events, the availability of this data provides unprecedented opportunities for urban scholars to depict brand new images out of the everyday familiar landscape of the urban setting.

*4.3. Big Data Supported Urban Studies/Science*

Through a meta-analysis of 48 urban big data studies, Wang and Yin [306] identified the essential qualities of urban big data. In a nutshell, urban big data focuses on refined spatiotemporal features and individual attributes at very fine levels (a street block, a building, etc.), and also has the capacity and impact to depict, predict, and manage cities through the complex interactions among individual data points and the collective trend such interactions demonstrate. This investigation agrees well with Batty [4] insightful observation that "cities are complex systems that mainly grow from the bottom up, their size and shape following well-defined scaling laws that result from intense competition for space." The emergence of urban big data provides a much-needed means to support the investigation of cities from the "bottom-up," and supplies a pathway to evaluate and investigate the scaling laws. An integrated urban theory is being gradually developed based on centuries of investigations of urban economics, urban land use, urban spatial and social structures, and urban transportation systems. Understanding the urban landscape and inherent urban growth dynamics requires indepth investigation facilitated by modern network science, allometric growth theory, and fractal geometry. With the arrival of mobile devices, the IoT stirred "urban big data" and infuses enormous information to facilitate the theoretical breakthrough of urban science as well as the socioeconomic environments of cities [295]. In the forum *Dialogues in Human Geography*, Batty [103] argues that the arrival of urban big data represents a sea change in understanding what happens where and when in cities. This is especially true with new methodological advancements for analyzing social sensing data for urban studies, such as temporal signature analysis, text analysis, and image analysis [307]. In addition, due to the dynamic characteristics of urban big data, it is shifting the emphasis of urban studies from longer term strategic planning to short-term thinking about how cities function and can be managed. This is evident in recently published big-data driven urban studies; see [54,114,115,117,122,308] for a few examples.

For instance, Huang and Wang [54] described four case studies using urban big data, including the detection of polycentric urban structures, evaluation of urban vibrancy, estimation of population exposure to $PM_{2.5}$, and urban land-use classification via deep learning. Their work suggests that urban big data plays a significant role in the sustainable planning of urban environments and reshapes traditional practices that often fall short in responding to everyday dynamics of the urban social, economic, environmental, spatial, infrastructure, and demographic landscapes.

Long-term planning, missions, and visions for urban development are critical for sustainable urban development, in both socioeconomic and environmental aspects. Long-term perspectives, however, are an averaged accumulation of short-term dynamics. The advent of urban big data and available means to acquire the data enable the in-depth

exploration and understanding of short-term dynamics of the everyday urban landscape. Studies of urban vibrancy have recently seen booming growth as a response to this change, which provides a chance for long-term planning to set a more practical goal based on everyday dynamics. In a recent study, Jia, Liu, Du, Huang and Fei [117] argue that urban vibrancy plays an important role in evaluating the quality of urban areas and guiding urban construction. The concept of urban vibrancy was proposed in 1961 by an American writer and urban activist, Jane Jacobs [309], in an attempt to oppose the then modernist urban planning efforts that overlooked and oversimplified the complexity of human lives in diverse communities within cities. In her mind, cities are prosperous, healthy, and sustainable only when their neighborhoods are vibrant and lively. Instead of intensive, large-scale, city-wide "renewal" or formulated planning practices, she valued urban vibrancy that originated from individual urban communities as an integrated part of a truly sustainable city. Her advocacy for dense mixed-use development and walkable streets has influenced later urban sustainable planning practices that focus on walkability and compact city spatial development in the US. The purpose of the vibrant planning idea is to bring "people" together instead of structured and formulated, grey, and impervious land uses that signify what cities used to be.

While the concept of urban vibrancy attracted urban planner and scholar attention since Jacobs' time, not surprisingly, putting the concept into action through actual planning practice took a long time to realize. This is because, while vibrancy is easy to define and observe, it is not easily quantified since we lacked consistent and continuous observation strategies and means. The booming of geotagged social media platforms and the widespread availability of handheld mobile devices (mobile phones, tablets, laptops, etc.) that can access these platforms provide excellent means to continuously and consistently observe how "people" gather and disperse in time and space in urban areas. He, Li, Liu, Wu, Zhang, Zhang, Liu and Yao [122] attempted an integrated approach that combines high resolution remote sensing images and real-time tencent user identity data and applied machine learning algorithms to quantify mixed-use development at the street level with the Shannon diversity index, which is often employed to measure species diversity in ecological studies and also applies to quantify mixed land use diversity in some recent studies [20,122,310]. Neighborhood vibrancy was found to be indeed correlated with the composite mixed land use measure.

In another study, Wang, Zhang, Yu, Qi and Li [114] collected Sina Weibo geotagged check-in data within a 24 h time span in the central district of Beijing and used it to proxy the urban vibrancy of Beijing in a full day's cycle. To investigate whether such representation provides any insightful guidance to urban planning and sustainable development, as well as to verify theories regarding Jane Jacobs [309] proposal of mixed-use development and compactness for urban vibrancy, they have employed a spatial autoregressive model and a multiscale geographically weighted regression model to explore potential determinants of urban vibrancy. Their results suggest that geotagged social media platform check-in data represents consistently and continuously people's activities over a day's cycle. Mixed-use land development, however, might not necessarily create a vibrant community, though different categories of infrastructure and facilities might. Their study further finds through spatiotemporal analysis that not only did urban vibrancy have a spatiotemporal variation pattern, the factors that influence urban vibrancy also vary across places and in different time spans. Apparently, the availability of vibrant urban status data through the big data scheme allows such varying patterns to be observed and results in a better, more practical, and sustainable urban future.

### 4.4. Big Data Facilitated Urban and Rural Integrated Development

In the recent trend of urban development, urban agglomeration [311–313] becomes a focus for many urban scholars. One of the key features within an urban agglomeration is the integrated development of urban centers and peripheral areas, including the rural area within the urban agglomeration [312]. In this trend of study, spatial big data plays an

increasingly important role in facilitating integrated development in both urban and rural areas. For instance, in 2010, Wang and Kilmartin [314] analyzed the call detail record data generated by mobile networks to reflect the dynamic behavior of humans across a range of temporal and spatial scales in Uganda. They examined the responses of subscribers to an economic incentive program regarding the mobile calling rate and identified distinctive patterns of rural and urban areas. More importantly, the analysis of the call detail record also reveals heightened economic activities in both urban and rural regions in Uganda. The approach reflects an objective spatial pattern that was naturally reflected in people's daily activities based on their economic status. In another study, Fang, Yu, Zhang, Fang and Liu [313] designed a web crawler to acquire 500,000 sets of geotagged Sina Weibo data in the Greater Beijing area (Beijing–Tianjin–Heibei) to study the spatial linkage between various places within the urban agglomeration. The results from analyzing the Sina Weibo data suggest a strong hierarchical structure existed within the urban agglomeration with the three cities (Beijing, Tianjin, and Shijiazhuang). The strongest linkage presents at the centers, however, the rural areas are loosely connected, even to the urban centers. They contended that the application of spatial big data reveals the need for more strategies to integrate urban and rural development for the healthy construction of vibrant urban agglomerations.

These studies provide a critical point of view when extending urban science to including not just cities, but the hinterlands that ultimately support city development. By integrating spatial big data with other forms of traditional data, such as demographic, economic, and environmental data, it is possible to gain a more comprehensive understanding of the interconnected systems that shape urban and rural areas. This information can inform decision-making and planning processes, leading to more informed and sustainable development strategies that promote integration between urban and rural areas. For example, spatial data can help identify transportation corridors and infrastructure needs that connect urban and rural areas [105,313,315], or it can inform strategies for preserving rural landscapes and promoting sustainable agriculture practices [316].

In a nutshell, the development of big data science and the advancement of more potential data sources can greatly benefit the investigation of rural changes and rural development within the context of urban science. By analyzing the flow of people, goods, and information between urban and rural areas through mining social media and internet search engines, big data science can provide a more comprehensive understanding of the interconnections between urban and rural areas and inform policies that promote integration and balance [317–319]. With the growing influence of urbanization on rural areas, exploring the potential of big data could provide insights into the effects of urbanization on rural communities and the environment, helping to guide policies and initiatives that promote sustainable rural development [299,320,321]. The importance of applying spatial big data in an integrated urban–rural development is also reflected in greater transparency and accountability in decision-making, ensuring that rural development policies are based on evidence and aligned with the needs and realities of rural areas, so that planning and policies can be derived to support more effective and sustainable urban–rural integrated development strategies that address the unique challenges and opportunities of rural areas under the context of urban science.

### 4.5. Limitations and Challenges of Applying Spatial Big Data in Urban Studies/Science

Conceptually, the availability and understandability of spatial big data, especially the ones acquired from social media platforms and global search engines, are easy to grasp. The meaning of such data and what it poses for urban science is also intriguing and informative. The hurdle is how to dig the stories out of the massive amount of information. With the increasing availability of spatial data from various sources, such as satellites, vehicle-bound sensors, social media, and search engines, the amount of data that needs to be processed and analyzed has grown exponentially. This requires significant computational resources and expertise, which can be a challenge for researchers with limited access to these resources or limited training in processing the data [132,322,323]. In addition, with the increased

amount of data, the need for appropriate data management and quality control is also increasing. Spatial data can be complex and often requires pre-processing and cleaning before it can be analyzed. This is a time-consuming and challenging task, particularly when dealing with data from multiple sources or when integrating data from different spatial scales as is often required in urban studies [121,122,324,325].

These challenges suggest that applying spatial big data analytical approaches in urban studies/science requires very careful planning and consideration prior to exploring the stories within the data. Being aware of these challenges and planning ahead of time with appropriate methods, techniques, and teamwork would be essential for taking the best advantage of what spatial big data could offer for the advancement of urban science.

## 5. Analyzing the New Data: Spatiotemporal Big Data Analytics and Bayesian Hierarchical Approaches

### 5.1. Spatial and Spatiotemporal Models

The availability of big data, the means to acquire the data, and the accumulation of high-resolution remote sensing images over urban areas, create unprecedented opportunities for urban scholars to evaluate existing urban theories and break grounds for new urban theories to be formulated. To achieve these goals, appropriate urban analytical methodologies are indispensable.

Urban models have long appeared in urban study literature to describe the formation, growth, and development of cities [1,4,103]. Early models focus on the spatial structure and land use patterns of cities, how such spatial structure and land use patterns changed over the years, and what factors might cause such changes with various traditional and newly developed machine learning algorithms [9,81,135,194,326–330]. Detection of the changes is often easily achieved, especially with the facilitation of GIS techniques and time series remote sensing images. Seeking mechanisms for such changes, however, is often a task that interests many urban scholars, since the answers provide direct guidance for better and sustainable urban planning and development.

Not surprisingly, regression analysis is among the most popular approaches since it attempts to establish a causal relationship between the outcome (changes in urban spatial structure and land use patterns, socioeconomic outcomes, demographic outcomes, etc.) and potential inputs [81,331,332]. The established relationships, if proven to be of significance, are the foundation for rational and practical urban planning. The availability of urban big data and urban remote sensing images enables urban scholars to have a more thorough view of everyday urban dynamics.

On the other hand, large amounts of data also suggest that extracting useful information from the sea of available information requires advanced analytical approaches that not only analyze assumed relationships, but also potential violations of modeling assumptions, added dimensions of space and time, and the complexity of dealing with urban big data and remote sensing information at a very fine scale [114,118,333,334].

One immediate violation of assumption when applying regression analysis to urban big data and remote sensing information is the existence of spatial autocorrelation among observed phenomena [335–339]. When analyzing data that is collected over geographic space, it is practically a norm to always examine potential spatial autocorrelation of the regression residuals to determine whether the assumed independent residual assumption is violated, and not surprisingly, it often is [219,332,340–344]. The remedy for such violation is usually the simultaneous autoregressive model proposed by Anselin [345], the spatial filtering techniques proposed by Griffith [346], or the nonlinear machine learning models as seen in Banihabib and Mousavi-Mirkalaei [347]. All these approaches have been proven to be effective in addressing the residual spatial autocorrelation issue. Elhorst [348] summarized that there are three potential interaction effects that could cause spatial autocorrelation in regression residuals, namely, the exogenous (spatial autocorrelation among the predictors), the endogenous (spatial autocorrelation among the outcome), and error or disturbance interaction (spatial autocorrelation among the residuals).

Due to the existence of spatial autocorrelation among the regression residuals (and we usually do not know how much autocorrelation there is—while autocorrelation can be detected and measured exploratorily, how much such autocorrelation is among different locations cannot be analytically determined), the commonly applied ordinary least squares estimation is no longer valid since the sum of the residuals is no longer well defined. The remedy is either the maximum likelihood estimator [348] or the spatial filtering regression [346]. For instance, Yu, Wei and Wu [219] and Yu and Wu [238] applied the maximum likelihood estimator in housing hedonic models in the city of Milwaukee. Yu, et al. [349] estimated sociodemographic factors' influence on tobacco outlet distribution in urban settings. Their conclusions, among many others, unanimously agreed that the maximum likelihood estimator not only performs better than the traditional nonspatial model but also reveals surprising results that were often masked if spatial autocorrelation was not taken into account. Similarly, using sociodemographic information, Yu, et al. [350] applied spatial filtering strategies to study China's urban attractivity, which produced a more refined understanding of the factors that influence individual migration tendencies among different cities in China.

A recent development in analyzing geospatial data is the addition of a temporal dimension, which enlarges the cross-sectional geographic data sets to become geopanel data sets. In his seminal book, Elhorst [348] provides a thorough examination of the spatial panel model's theoretical, methodological, and estimation aspects. While dealing with the added temporal dimension, the spatial filtering approach often has to rely on a random effect model due to the mixture of individual effects and spatial effects during estimation [351].

Apart from the regression residuals' spatial autocorrelation problem, the spatial nonstationarity issue [352] that commonly existed in the regressed relationships among data collected over geographic spaces is another major concern in analyzing large amounts of remotely sensed image data and social sensing acquired "big data." Spatial nonstationarity has been observed as a norm in geographic datasets, and the initial treatment is brilliantly expressed in the geographically weighted regression model [352,353], with modifications from the spatial filter supported by the varying coefficient model [354]. The addition of a temporal dimension to the varying coefficient model was proposed in [355,356] within the geographically weighted regression framework and also discussed in the spatial filtering literature [351,354,357–359]. A recent work that applies multiscale geographically weighted regression analysis with remote sensing and big data on urban vibrancy in Beijing [114] suggests spatiotemporal models provide insightful conclusions to urban planners and governance.

*5.2. Bayesian Hierarchical Spatiotemporal Analysis*

While admittedly, the spatial and spatiotemporal models in the frequentist statistical analysis framework have produced many insightful discussions in urban studies and the utilization of urban remote sensing and big data [114–121,360–362], it was often found that while data sets are abundant, our experiences and prior knowledge were also enhanced dramatically in the remote sensing and big data era. Quantification of experiences and prior knowledge will be tricky under the frequentist statistical analysis framework, but this is one of the strengths of Bayesian statistical analysis. This is especially true when applying analytical approaches to data that is collected over geographic and spatiotemporal spaces.

The "First Law of Geography" proposed by Tobler [363] summarized nicely the commonly observed distribution of geographical objects and their attributes that "everything is related to everything else, but near things are more related than distant things." Whether or not this is indeed a "Law" that existed among geographic or spatiotemporal observations, or simply a spatial variant of post hoc fallacy [364], the observed spatial autocorrelation is not only the foundation for frequentist spatial and spatiotemporal data analysis but also an excellent premise for accumulated experience and the prior knowledge of data sets that are collected over geographic and spatiotemporal spaces.

Moreover, while data is collected over geographic and temporal spaces, because of the inherent "autocorrelation" among spatial/spatiotemporal neighbors, we can apply a hierarchical modeling framework to study both the individual data-generating process at each spatial/spatiotemporal location and the interactive data generating processes among neighbors [365]. Based on the Gaussian Markov Random Field theory [366], Rue and colleagues [366–368] unified the spatial and spatiotemporal models in the frequentist framework under the Bayesian hierarchical modeling framework [365,369,370].

Without specifying the geographic, temporal, or spatiotemporal context in Rue and Held [366], a Gaussian Markov Random Field is a Gaussian random field $x \sim N(\mu, \Sigma)$ that satisfies:

$$p\big(x_i \big| \{x_j : j \neq i\}\big) = p\big(x_i \big| \{x_j : j \in \mathcal{N}_i\}\big) \tag{1}$$

where $\mathcal{N}_i$ is a set containing all the neighbors of observation $i$. It is further found that the precision matrix $Q$ of the random field, which is the inverse of the covariance matrix $\Sigma$, contains only nonzero values for neighbors and the diagonal elements. Neighborhood structures are inherent in geospatial and spatiotemporal data-generating processes; hence the geographic and spatiotemporal processes are naturally Gaussian Markov Random Fields. The specific characteristics of the precision matrix $Q$ for a Gaussian Markov Random Field enable direct calculation and simulation for Bayesian analysis, especially considering the neighborhood structure among geographic and spatiotemporal observations has been well studied in geospatial analytics. Priors that satisfy the autocorrelation neighborhood structure can be specified with different levels of confidence and used for modeling purposes. The results from the models can then be compared with reality and further refine the priors for better and more accurate modeling.

A few studies attempting to apply the spatially varying coefficient model to study urban crime and influencing factors under the Bayesian hierarchical modeling framework are now emerging [134,371], demonstrating the potential of coupling the advanced Bayesian hierarchical spatial/spatiotemporal modeling framework with available remote sensing and spatial big data in urban settings. The incorporation of previous knowledge through the defined prior about the existing, but usually not exactly known, spatial/spatiotemporal autocorrelation [371] among observations collected over geographic and spatiotemporal spaces, provides a rather robust and intuitive way to integrate this inherent characteristic of the data into the analysis. In this regard, Bayesian hierarchical spatial–spatiotemporal data analysis is more sensible and flexible when dealing with complex interactions among observed data, especially when the amount of data is large, yet previous knowledge is abundant, as is typical in many urban social, economic, environmental, ecological, demographic, and morphological studies, among many others.

## 6. Future Research Direction

This review covers the development of urban studies/science, applications of emerging remote sensing and big data techniques in urban studies, and an introduction to how to best analyze ever-increasing remote sensing and spatial big data narratives in urban settings. It is not the authors' intention to provide a comprehensive and all-inclusive review of these constantly developing fields. Instead, this review attempts to serve the purpose of attracting more insightful and powerful analyses and understanding of the highly dynamic, highly complex, highly interactive, and highly vibrant urban spaces of today's globalized world. While the internet began spreading across the globe, and advanced transportation technologies seemed to bring the Earth together (terms like Earth Village still sound fresh even now), and space and congregation of people seemed to be becoming a thing of the past, the last three decades suggest that human beings are still congregative beings and cities are still among the most attractive locations for the majority of us. The world's population just went past eight billion people and, in the foreseeable future, over half of these eight billion people will likely still live in cities of various sizes, occupying only a fraction of the vast surface of our Earth. Utilizing the newly emerged abundant remote sensing and spatial big data to their fullest potential becomes imperative for a better, more sustainable

urban future. The future of urban studies/science will increasingly rely on the openness of accumulated "big data" on pressing issues such as air quality [372], environmental degradation, real-time traffic flows, and the ebbs and flows of various human activities. How the average citizen can access and interact with these data sets either through direct participation [373] or crowdsourcing [374,375] is key to a livable and sustainable urban future. This current review serves the purpose of bringing together remote sensing, spatial big data, and spatial–spatiotemporal analytical methods to advance urban science in the new globalization and digital era.

During our reviewing process, sorting through the many studies that we investigated, we sense the field of urban science is constantly evolving, and new trends and challenges are emerging as scholars explore the potential of remote sensing and spatial big data analysis for understanding and managing urban environments, urban land use strategies, urban heat island effect, and many other existing and emerging urban issues. With the increasing integration of remote sensing and spatial big data with urban science, we envision a few potential trends and challenges for future urban science and studies.

First, urban science will rely more on data integrated from multiple sources. The availability of spatial data from multiple sources is increasing rapidly, and future research will need to focus on integrating data from these diverse sources, including satellites, mobile devices, and social media. This will require the development of new methods and tools for data integration, as well as a greater understanding of the strengths and limitations of different data sources.

Second, urban science will increasingly integrate machine learning and artificial intelligence techniques in its investigation. The use of machine learning and artificial intelligence in urban science is expected to increase in the future with the tighter integration of remote sensing and spatial big data into urban studies, as researchers seek to automate and streamline the analysis of large datasets. This will require the development of new algorithms and models that can effectively process and analyze spatial data, as well as a greater understanding of the ethical implications of these technologies.

Third, urban science will tap into higher spatial and temporal resolution data to suit its own research agenda. The recent development of new satellite and sensor technologies is leading to the availability of high spatial and temporal resolution data [124,154], which will enable urban scholars to study urban environments in greater detail than ever before. However, this will also require the development of new methods and tools for processing and analyzing this data, as well as a greater understanding of the limitations of these technologies.

Fourth, urban policies will increasingly rely on big data analytics [114,135,324,376]. The use of spatial big data analysis for urban policy is becoming important, as decision-makers seek to use data-driven approaches to address urban challenges such as climate change [155], transportation [377,378], and public health [133,379,380]. Future research will need to focus on developing new analytical methods and tools that can effectively inform urban policy and decision making. This became evident during the recent COVID-19 pandemic [381].

Fifth, integrating remote sensing and especially spatial big data that is based on social sensing with urban science will also pay particular attention to potential ethics and privacy concerns. For instance, as new technologies such as facial recognition and geolocation become more widely available, urban scholars will need to address these concerns and develop ethical frameworks and guidelines for the use of spatial data in the urban science research framework.

Without a doubt, the future of integrating remote sensing and spatial big data analysis with urban science is likely to be characterized by rapid advances in technology, the development of new analytical methods and tools, and a greater emphasis on ethical and privacy concerns. Addressing these challenges will require interdisciplinary collaboration and a greater understanding of the complex social, economic, and environmental factors that shape urban environments.

## 7. Conclusions

Through the review of the integrated contribution of remote sensing, spatial big data, and advanced spatial–spatiotemporal analytical methods in urban studies/science, our review provides a comprehensive framework for an integrated understanding of urban science development in the new century. This literature review suggests urban studies and urban science have benefited enormously from the development of remote sensing technologies and the availability of spatial big data. The emergence of advanced and sophisticated spatial–spatiotemporal analytical methods contributed to the further development of urban science with these newly available urban data sources.

Through this review, we see that detailed, almost real-time availability of high spatial and temporal resolution remote sensing data and spatial big data opens the door to understanding the ultimate dynamics of the cityscapes, prompting practical research themes under the everyday urbanism theoretical framework [110]. We are now able to monitor urban environment events, urban population changes, traffic patterns, pedestrian flows, business operations, and other vibrant urban characteristics almost in real time. This information is accumulatively stored in the cloud for better urban management. Advanced analytical methods allow scholars to delve into this sea volume of information and reveal the detailed urban dynamics behind it. This is a luxury that preremote sensing and spatial big data era urban studies would have dreamed of. There is no doubt that new data and new methodologies bring new theoretical breakthroughs. A better and on-time understanding of urban dynamics leads to better urban planning and development, which facilitates the virtuous cycle between urban dwellers and the urban environments, leading to a sustainable urban future.

We will see more of the new types of information emerging in the future. However, as of right now, our mission remains steadfast to work with the accumulative data sets from remote sensors, social sensors, and other geotagged data collecting platforms to reveal the dynamic urban landscape, boost the ever-changing urban vibrancy, and guide the future development of this most congregative space to be sustainable and prosperous.

**Author Contributions:** Conceptualization, D.Y. and C.F.; methodology, D.Y.; validation, D.Y. and C.F.; investigation, D.Y.; resources, D.Y. and C.F; writing—original draft preparation, D.Y; writing—review and editing, D.Y.; visualization, D.Y.; supervision, D.Y. and C.F.; project administration, D.Y.; funding acquisition, D.Y. and C.F. All authors have read and agreed to the published version of the manuscript.

**Funding:** This research was supported by the Chinese National Nature Science Foundation, grant number 42121001 (C.F).

**Data Availability Statement:** Not applicable.

**Conflicts of Interest:** The authors declare no conflict of interest.

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
