# Peer review of "Urban Remote Sensing with Spatial Big Data: A Review and Renewed Perspective of Urban Studies in Recent Decades"

_remotesensing, doi:10.3390/rs15051307_

Round 1

Reviewer 1 Report (Previous Reviewer 1)

This manuscript has not been sufficiently improved to warrant publication in Remote Sensing. Though this paper has reviewed more than 300 references, it still does not meet the standards of this journal.  

The authors must summarize the representative methods of each direction, and conduct experiments to make  comparative analysis of the representative methods, not just a simple summary of words. Currently, I cannot find any valuable contributions. This is a high-level SCI journal, even if the review paper, also must have their own substantial contributions, it is suggested to submitted to other related journals.

Author Response

Response: Thank you again for your critical review. While it is very tempting to “summarize the representative methods of each direction,” as the manuscript stands right now, the task might very well be divided into separate review articles. We do appreciate your critical thinking, but “conducting experiments to make comparative analysis of the representative methods” might be beyond the scope of a literature review article.

While we cited over 300 references (close to 400), but our review has gone far beyond that number. The summary of words in the current review attempts to comment and compare studies of urban remote sensing over the past 70 plus years, and urban spatial big data over the past 20 plus years. We have done our comparative analysis through the thorough exploration of these valuable research and summarize their results for an organized study. We sincerely hope you could provide more specific criticism on what exactly you might refer to “make comparative analysis of the representative methods” so that we can improve our manuscript following your critical comments.

In any cases, we strive to make our review as high-caliber as we could prepare and write for. As we are very well aware of the prestigiousness of the journal Remote Sensing.

Reviewer 2 Report (Previous Reviewer 3)

The manuscript has huge improvements, and the authors have now addressed all comments that I have raised. 

Nevertheless, the title of this manuscript should be changed into

"Urban remote sensing with spatial big data: A review and renewed perspective of urban studies in recent decades"

Other than that, I am happy to recommend the publication of this manuscript.

Author Response

Response: Thank you very much for your suggestion and meticulous review. Per your suggestion, we have changed the title to “Urban remote sensing with spatial big data: A review and renewed perspective of urban studies in recent decades.”

Reviewer 3 Report (New Reviewer)

Thank you for giving me this opportunity to read the manuscript entitled "Urban remote sensing with spatial big data: a literature review and renewed aspect of urban studies in the new century".The paper provides an in-depth review of how remote sensing data sources and spatial big data analytical technologies are being used in urban studies.

The authors provide an overview of the various data sources and technologies that are available for urban studies and discuss how they are being used to produce realistic and valuable results for urban planners and governments. The review is well-organized, well-written, and provides a useful guide for researchers interested in the topic. The topic of this manuscript is interesting and would be a good contribution to this field. I think it could be considered for publication in RS once the following issues are addressed.

1. Please replace the keywords that already appear in the manuscript's title with close synonyms or other keywords, which will also facilitate your paper being searched by potential readers.

2. The authors should expand on the limitations and challenges of remote sensing data sources and spatial big data analytical technologies. This will provide readers with a more nuanced understanding of the topic.

3. Line 263 “… urban water body/green space extraction and mapping [166-171]”: papers titled “How does urban expansion impact people’s exposure to green environments? A comparative study of 290 Chinese cities” and “Observed inequality in urban greenspace exposure in China” are suggested to be cited to support the statement here.

4. Some grammatical errors exist in the manuscript. Therefore, a critical review of the manuscript's language will improve its readability."

Author Response

Thank you for giving me this opportunity to read the manuscript entitled "Urban remote sensing with spatial big data: a literature review and renewed aspect of urban studies in the new century". The paper provides an in-depth review of how remote sensing data sources and spatial big data analytical technologies are being used in urban studies.
The authors provide an overview of the various data sources and technologies that are available for urban studies and discuss how they are being used to produce realistic and valuable results for urban planners and governments. The review is well-organized, well-written, and provides a useful guide for researchers interested in the topic. The topic of this manuscript is interesting and would be a good contribution to this field. I think it could be considered for publication in RS once the following issues are addressed.
Response: Thank you for your agreement and meticulous review. We strive to address your concerns and hope to gain your further support.

1. Please replace the keywords that already appear in the manuscript's title with close synonyms or other keywords, which will also facilitate your paper being searched by potential readers.
Response: Thank you for this comment. We have replaced the keywords “meta-analysis” to “literature study,” “Remote Sensing” to “urban remote sensing,” “spatial big data” to “urban spatial big data” to reflect the key ideas of the review and facilitate potential readers’ search. We hope the change will gain your support.

2. The authors should expand on the limitations and challenges of remote sensing data sources and spatial big data analytical technologies. This will provide readers with a more nuanced understanding of the topic.
Response: This is a very productive suggestion. We have added two paragraphs at the end of the newly organized sections 3 and 4 that specifically review the limitations and challenges of applying remote sensing and spatial big data in urban science. We hope this addition will gain your support.

3. Line 263 “… urban water body/green space extraction and mapping [166-171]”: papers titled “How does urban expansion impact people’s exposure to green environments? A comparative study of 290 Chinese cities” and “Observed inequality in urban greenspace exposure in China” are suggested to be cited to support the statement here.
Response: Thank you very much for supplying with useful and relevant work here. After careful review of the manuscripts your suggested, we found them to be closely related to the discussion here and we have included the references in the literature.

4. Some grammatical errors exist in the manuscript. Therefore, a critical review of the manuscript's language will improve its readability."
Response: we again asked a native speaker to carefully review through the manuscript to correct any possible grammatic error and wording issue. We hope the revision would gain your support.

Reviewer 4 Report (New Reviewer)

I want to congratulate the authors for their efforts in this manuscript. They have conducted an exhaustive bibliographic review. There are some aspects which can be improved in order to facilitate its lecture to readers. Moreover, a short subsection describing the methodology conducted to collect the data must be included (used engine, keywords, years…). Following, I add some recommendations to improve the reader-friendly aspect of the paper:

-If possible, add Tables and Figures. It is easy to create graphics including the number of published papers per year and even include one in each section to show the reader when appeared the included topics in the paper. Tables can be used to summarize the data described in the text.

-In the methodology subsection, besides detailing how the papers were searched, compare the survey with other ones. Thus, it makes it possible to highlight the novelty of this survey compared with other surveys which review remote sensing (10.1016/j.isprsjprs.2015.10.004, https://doi.org/10.3390/app12168045, 10.1016/j.isprsjprs.2019.11.023, 10.1016/j.jag.2021.102514)

-Section 3 is exceptionally long. Check if it is possible to split it into different sections. In general terms, surveys might include a large number of sections.

-Consider using bullet points or subsections when describing the Future Research Direction to split it into diverse topics (challenges/trends).

-Include the conclusion in an independent subsection.

Author Response

I want to congratulate the authors for their efforts in this manuscript. They have conducted an exhaustive bibliographic review. There are some aspects which can be improved in order to facilitate its lecture to readers. Moreover, a short subsection describing the methodology conducted to collect the data must be included (used engine, keywords, years…). Following, I add some recommendations to improve the reader-friendly aspect of the paper:

Response: Thank you for your encouragement and critical comments. We will try our best to address your concerns fully. We have added a methodology subsection within the introduction section detailing how the data for our literature review is collected.

-If possible, add Tables and Figures. It is easy to create graphics including the number of published papers per year and even include one in each section to show the reader when appeared the included topics in the paper. Tables can be used to summarize the data described in the text.

Response: Thank you for your suggestion. We have figure 1 to summarize the literature we searched. Our current review is already exceedingly long, while we admit that tables and figures will be informative to readers, under the current structure and length of the manuscript, adding additional figures and tables might not be particularly productive. We do appreciate your suggestion, though, but hope our decision of not adding additional tables and figures will gain your approval and support.

-In the methodology subsection, besides detailing how the papers were searched, compare the survey with other ones. Thus, it makes it possible to highlight the novelty of this survey compared with other surveys which review remote sensing (10.1016/j.isprsjprs.2015.10.004, https://doi.org/10.3390/app12168045, 10.1016/j.isprsjprs.2019.11.023, 10.1016/j.jag.2021.102514)

Response: Thank you for supplying these valuable reviews of remote sensing so that we can have a more comprehensive review. We have added relevant comparison in the revision and hope the revision will gain your support.

-Section 3 is exceptionally long. Check if it is possible to split it into different sections. In general terms, surveys might include a large number of sections.

Response: We have separated section 3 to two sections: remote sensing and how it advances urban science; spatial big data and how it advanced urban science. We hope this structural change will make the organization of the manuscript clearer.

-Consider using bullet points or subsections when describing the Future Research Direction to split it into diverse topics (challenges/trends).

-Include the conclusion in an independent subsection.

 Response: We are now separating the conclusion and future research direction into two separate sections, and conclusion is its own section. We have gone back to our literature search and summarized five potential future trends and challenges in the future research direction section. We hope this revision will make the structure of the manuscript cleaner and hope to gain your support.

Round 2

Reviewer 4 Report (New Reviewer)

Authors have addressed the mentioned issues. We accept the explained motivation for not including more figures requested in comment 1.

This manuscript is a resubmission of an earlier submission. The following is a list of the peer review reports and author responses from that submission.

Round 1

Reviewer 1 Report

The contributions of this paper are limited and the state-of-the-art compared methods are needed. This review contains many inappropriate language and grammatical expressions, and the readability of the article is quite problematic. The presentation of the paper is not clear enough and needs to be sorted out more carefully. The data used, the representative methods and the comparison of experimental results have been not discussed. Additionally, the biggest problem is the hollowness of the summary as a review, which has low referentiality for readers. It is difficult to find an obvious innovation and important contribution in this work. Thus, this work does not meet the standard of Remote Sensing.

Author Response

The contributions of this paper are limited and the state-of-the-art compared methods are needed. This review contains many inappropriate language and grammatical expressions, and the readability of the article is quite problematic. The presentation of the paper is not clear enough and needs to be sorted out more carefully. The data used, the representative methods and the comparison of experimental results have been not discussed. Additionally, the biggest problem is the hollowness of the summary as a review, which has low referentiality for readers. It is difficult to find an obvious innovation and important contribution in this work. Thus, this work does not meet the standard of Remote Sensing.

Response: Thank you for your critical comments. We have asked a native English speaker to go through the manuscript intensively to correct any language and grammatical issues to increase the readability of the article.

This is a review article and we have extensively reviewed the development of urban studies, remote sensing and social sensing based spatial big data and how the latter two support the development of urban studies. We have compared from a relatively broad and holistic perspective the theoretical arguments, data sources, and methodological advancements that are involved in applying remote sensing and spatial big data to urban studies and urban science. Since this is a review article, we did not conduct any experiments or apply any specific methods or analyze any specific data. We have tried our best to fill in the hollowness of the summary in this revision to enrich the review’s contribution to the remote sensing scholarly community. We hope the overall revision will gain your approval and we appreciate your comments for making the manuscript a better contribution.

Reviewer 2 Report

In this revision, the authors have adequately responded to the comments and suggestions of the reviewers. It is now acceptable for publication.

Author Response

Thank you for your support. We appreciate your review and help.

Reviewer 3 Report

This review article explores the historical development of spatial and temporal big data, the use of data science and analytical approaches in urban and environmental studies, as well as in modern planning and predicting changes in urban landscapes. The topic itself is of practical interests and the coverage is very broad, and the review is also a good summary of all these relevant topics. There are some major and minor edits needed as follows:

Major Change needed / Comments

(1) The review as a whole focus on urban areas, however it would be good to include a sub-section near the end of this review, on how big data science and advancement of more potential data sources could benefit the investigation of rural changes and rural development in recent decades.

(2) In Section 1 (Introduction), the authors should also highlight the relationship and connection between changing land use patterns with city development, the rural-to-urban transition, especially in data limited region / developing cities. Some good references are as follows:

https://www.mdpi.com/2072-4292/13/16/3337

https://www.mdpi.com/2073-445X/11/2/313

https://ecologicalprocesses.springeropen.com/articles/10.1186/s13717-016-0044-6

(3) Lines 78-79: What do you mean by "system perspective"? Please kindly explain the concept.

(4) Lines 94-95: The authors cited a reference from Koh et al. [127] here, on urban public health, however this topic is rather far away from the focus of current review report, therefore relevant sentences and references should be removed from the current manuscript.

(5) Line 104: "the current review aims to fill in this gap" - please give some description of the gap.

(6) Lines 107-115: Suggest to remove this paragraph, or just highlight the key point will already be fine.

(7) Lines 128-134: The authors mentioned the use of Bayesian framework, but why is such framework better than machine learning and data assimilation approaches? More recent research adopt ML methods.

(8) Lines 147-160: The historical development of "cities" and "human beings" should be simplified. Some references should also be removed too.

(9) Lines 170-174: The authors can give some more specific examples (from geographical point of view).

(10) Lines 181-195: The authors mentioned that many models relied on simple geometric explorations under ideal conditions, however nowadays models could easily incorporate land use patterns / terrain of a spatial region as its inputs, therefore the problem mentioned here will not be a big matter of concern. Please try rewriting this sub-paragraph.

(11) Section 3 (Lines 206-232) - for the application of remote sensing in urban science, the authors should also focus on environmental detection and change of environmental conditions, for example air quality assessments, identification of pollution hotspots via satellite means. Some references are as follows:

https://journals.ametsoc.org/view/journals/bams/101/1/bams-d-18-0013.1.xml

https://ui.adsabs.harvard.edu/abs/2017AGUFM.A53A2206L/abstract

https://www.mdpi.com/2072-4292/10/11/1789

https://www.mdpi.com/2072-4292/12/22/3803

(12) Lines 266-269: The authors should highlight and describe the advancement of spatial / temporal resolution of Landsat images throughout recent decades too.

(13) Lines 290-295: Should include relevant references here, and to explain the deficiency occurred in developing countries.

(14) Lines 346-347: What is the rationale / reason for estimating population figures?

(15) Line 383: The motivation of using nighttime light data in urban studies should be explained in advance.

(16) Lines 406-409: Any studies that connect the use of "nighttime light" data and "anthropogenic activities"? Some references should be added here.

(17) Lines 420-426: The authors can connect the use of these available platforms and social network / media with COVID database / medical platforms invented during recent pandemic.

(18) Lines 440-442: The authors can write a bit more on the advancement of computing techniques and database system, and definitely, please include some historical documents as citations.

(19) Lines 452-455: The connection of these information with remote sensing might not be that clear. 

(20) Lines 488-501; Lines 511-537: These several paragraphs can be shortened, and should focus more on clustered computation.

(21) Lines 576-579: Here, the authors should focus more on techniques adopted / applied into relevant investigation of urban science and spatial analytics.

(22) Lines 608-613: Although impervious land uses indicate city development, they may serve for different land use / purpose in rural regions too, for example, in some counties / villages of Japan. Please refer to the architectural style and rural/urban design of Japan, these remote areas can still be walkable.

(23) Section 5: After mentioning Lines 797-805, the authors should talk more about data openness, data integration, for example, data openness of air quality, environmental datasets, or sharing of spatial attributes / dashboard among countries / cities ; as well as how public engagement could lead to good practices of promoting urban development, sustainability and city planning.

(24) Line 802: "extending the theoretical framework of everyday urbanism" - the idea is important, however it is not very properly highlighted in the entire manuscript. Suggest to rephrase the idea here.

Minor changes / typos:

Line 17: Urban scholars are now equipped

Line 22: reviews the development, current status, and future...

Figure 1: Font size should be larger.

Figure 2: "Big data thinking" - should it be more specific like "use of big data"?

Lines 135-145: The authors should also mention the concept of "data integration" in this context.

Line 184: Infrastructural constructions such as...

Line 190: For "air traffic", do you mean "flights"?

Line 194: Please change the phrase "well the then"?

Line 241: "other fields" - what are those fields?

Line 243-244: Please remove "the then"

Lines 245-247: It actually depends on the goal and the scale actually, for example, meso-scale and micro-scale applications, all these scales could be feasible.

Line 315: The authors should describe the distinct dimensions of urban forms.

Line 318: critical to support the 

Line 328: satellite-based high resolution images

Line 472: dataset that

Line 556: reasonably inferential

Line 588: the "2.5" of "PM2.5" should be under-scripted.

Line 589: plays a significant role...

Line 594: development, in both socioeconomic and environmental aspects

Line 624: Please describe the use and main idea of this "Shannon diversity index"

Lines 632-633: What are these spatial and spatio-temporal analytical strategies?

Line 674: immediate violation of assumption

Line 692: no longer well-defined.

Line 698: better than the traditional

Line 699: that were often masked

Line 701: urban attractivity, which produced...

Line 713: remotely sensed image data and...

Line 721: How do you explain / define the word "advanced"?

Lines 734-735: that everything in world is inter-connected, but...

Line 743: process at each

Line 758: enables direct calculation and simulation for Bayesian analysis, especially

Line 762: compared with the reality

Line 764: remove "an"

Lines 762-763: Do the authors mean "data assimilation" here?

Generally speaking, the manuscript is of high impact, and the ideas and concepts are explained in a very established manner. The authors have done a lot of work before writing this review article. After addressing all the above comments and suggestions / changes, I am sure the review will be much better.

Author Response

This review article explores the historical development of spatial and temporal big data, the use of data science and analytical approaches in urban and environmental studies, as well as in modern planning and predicting changes in urban landscapes. The topic itself is of practical interests and the coverage is very broad, and the review is also a good summary of all these relevant topics. There are some major and minor edits needed as follows:

Response: We thank you for your insightful and encouraging comments. We will try our best to address your concerns to the fullest.

Major Change needed / Comments

(1) The review as a whole focus on urban areas, however it would be good to include a sub-section near the end of this review, on how big data science and advancement of more potential data sources could benefit the investigation of rural changes and rural development in recent decades.

Response: Thank you for your comment.

(2) In Section 1 (Introduction), the authors should also highlight the relationship and connection between changing land use patterns with city development, the rural-to-urban transition, especially in data limited region / developing cities. Some good references are as follows:

https://www.mdpi.com/2072-4292/13/16/3337

https://www.mdpi.com/2073-445X/11/2/313

https://ecologicalprocesses.springeropen.com/articles/10.1186/s13717-016-0044-6

Response: Thank you for this insightful comment. We have gone through the references you listed and highlighted the relationship and connection between changing land use patterns with city development, the rural-to-urban transition, focusing on the discussion on how the newly emerging data source would benefiting the data limited region and developing cities.

(3) Lines 78-79: What do you mean by "system perspective"? Please kindly explain the concept.

Response: Thank you for your comment. We termed “system perspective” as treating the entire place as an integrated system so that different aspects of the place can be expressed as different components that interact with one another. We clarified the term in this revision and hope this clarification will gain your approval and support.

(4) Lines 94-95: The authors cited a reference from Koh et al. [127] here, on urban public health, however this topic is rather far away from the focus of current review report, therefore relevant sentences and references should be removed from the current manuscript.

Response: Thank you for your meticulous review. We further reviewed the relevance of Koh’s work with our review, and we agree with your comments. We have removed relevant expressions in our current revision and hope the revision will gain your support.

(5) Line 104: "the current review aims to fill in this gap" - please give some description of the gap.

Response: Thank you for your critical comments. This sentence shall be immediately following the last paragraph when we stated that “A comprehensive review of how remote sensing and spatial big data, as well as spatial/spatiotemporal research methodology influence the research of urban science as a whole, however, merits further investigation.” We connect these two paragraphs together to provide a more coherent expression. We hope this revision will gain your approval and support.

(6) Lines 107-115: Suggest to remove this paragraph, or just highlight the key point will already be fine.

Response: We dropped the paragraph as you suggested but only left a few key point to avoid redundancy. Thank you for your careful review.

(7) Lines 128-134: The authors mentioned the use of Bayesian framework, but why is such framework better than machine learning and data assimilation approaches? More recent research adopt ML methods.

Response: Thank you for your critical comment here – we do not intend to compare Bayesian framework with recent machine learning and data assimilation approaches. Instead, the introduction of Bayesian framework here focuses on its capability to deal with spatial and spatiotemporal data that can take advantage of the traditional geographic knowledge (spatial autocorrelation). We do not intend to cover all newly developed machine learning methods, nor do we have the capacity to do so. We do, however, as you suggested, add comments about how the recent development of machine learning algorithms, such as the Random Forest, Support Vector Machine, Neural Network approaches are of great importance for the application of remote sensing and spatial big data to urban studies and science during the later sections when we give specific review of the Bayesian framework.

(8) Lines 147-160: The historical development of "cities" and "human beings" should be simplified. Some references should also be removed too.

Response: We followed your suggestion and simplified this entire section and removed less relevant references.

(9) Lines 170-174: The authors can give some more specific examples (from geographical point o,f view).

Response: we have presented more specific examples that apply the central place theory to study urban hierarchical structures here.

(10) Lines 181-195: The authors mentioned that many models relied on simple geometric explorations under ideal conditions, however nowadays models could easily incorporate land use patterns / terrain of a spatial region as its inputs, therefore the problem mentioned here will not be a big matter of concern. Please try rewriting this sub-paragraph.

Response: thank you for this critical comment. We have restructure the paragraph to emphasize that the limitation applies only to the early stages of urban studies because of limited computational power and observational data sources at the time.

(11) Section 3 (Lines 206-232) - for the application of remote sensing in urban science, the authors should also focus on environmental detection and change of environmental conditions, for example air quality assessments, identification of pollution hotspots via satellite means. Some references are as follows:

https://journals.ametsoc.org/view/journals/bams/101/1/bams-d-18-0013.1.xml

https://ui.adsabs.harvard.edu/abs/2017AGUFM.A53A2206L/abstract

https://www.mdpi.com/2072-4292/10/11/1789

https://www.mdpi.com/2072-4292/12/22/3803

Response: Thank you for this constructive suggestion. Lines 206-232 are a general prelude for late discussion of remote sensing and spatial big data. We have added relevant narratives and references in section 3.1.1 Extracting and analyzing physical environments of urban areas. We hope the modification would gain your approval.

(12) Lines 266-269: The authors should highlight and describe the advancement of spatial / temporal resolution of Landsat images throughout recent decades too.

Response: Thank you for this comment. We have included the recent advancement of the spatial /temporal resolution of Landsat images in this section.

(13) Lines 290-295: Should include relevant references here, and to explain the deficiency occurred in developing countries.

Response: Thank you for the comment – the primary deficiency occurred in developing, especially countries in the sub-Sahara African region is the lack of financial, technological, and human capital resources. We clarified this deficiency in the revised version. We hope the revision will gain your support.

(14) Lines 346-347: What is the rationale / reason for estimating population figures?

Response: thank you for the comments. We clarified that using remote sensing images to estimate population figures is a relatively convenient and cost-effective approach compared to a full-scale census or even a 1% or 5% demographic survey (such as the American Community Survey conducted annually). We added this clarification to the revision and hope to gain your support.

(15) Line 383: The motivation of using nighttime light data in urban studies should be explained in advance.

Response: Thank you for your suggestion. We have provided the rationale for using nighttime light data in advance prior to further discussion. We stated that “This is because the intensity of various modern human activities in a place is closely related with the amount of energy consumed there. Nighttime light emission provides an immediate proxy for the intensity of energy consumption, hence a good proxy for a wide variety of human socioeconomic activities.” We added relevant citations to support the argument here as well. We hope the revision will gain your support.

(16) Lines 406-409: Any studies that connect the use of "nighttime light" data and "anthropogenic activities"? Some references should be added here.

Response: Thank you for the comments. We have added a few relevant citations there to support our argument. We hope the revision will gain your support.

(17) Lines 420-426: The authors can connect the use of these available platforms and social network / media with COVID database / medical platforms invented during recent pandemic.

Response: We have added a relevant narrative regarding the availability of social media platforms and the COVID database during recent pandemic to highlight the importance of social media data generation. We hope this addition will gain your support.

(18) Lines 440-442: The authors can write a bit more on the advancement of computing techniques and database system, and definitely, please include some historical documents as citations.

Response: Thank you for your comments, we have added some further examples about social mining and search engine mining in this section. We hope this revision will gain your approval.

(19) Lines 452-455: The connection of these information with remote sensing might not be that clear. 

Response: we have softened the tone here to reflect the relatively unclear connection between social sensing and remote sensing. We hope the modification will gain your approval and support.

(20) Lines 488-501; Lines 511-537: These several paragraphs can be shortened, and should focus more on clustered computation.

Response: we have shortened these two places to provide more concentrated discussion of big data and big data era.

(21) Lines 576-579: Here, the authors should focus more on techniques adopted / applied into relevant investigation of urban science and spatial analytics.

Response: we modified the statement here to reflect more on how emergence of urban big data facilitates and supports the study of cities from “bottom-up” and further develop urban theories in the big data era. We hope this revision will gain your approval. Thank you for your insightful suggestion.

(22) Lines 608-613: Although impervious land uses indicate city development, they may serve for different land use / purpose in rural regions too, for example, in some counties / villages of Japan. Please refer to the architectural style and rural/urban design of Japan, these remote areas can still be walkable.

Response: Thank you for this critical comment. We have added a geographic confinement to avoid unnecessary confusions here.

(23) Section 5: After mentioning Lines 797-805, the authors should talk more about data openness, data integration, for example, data openness of air quality, environmental datasets, or sharing of spatial attributes / dashboard among countries / cities ; as well as how public engagement could lead to good practices of promoting urban development, sustainability and city planning.

Response: Thank you for your suggestion. We followed the suggestion and provided more information about the future of urban studies/science and how it will rely on the openness of accumulated big data, and the importance of public engagement in interacting with the information. We hope the revision will gain your approval and support.

(24) Line 802: "extending the theoretical framework of everyday urbanism" - the idea is important, however it is not very properly highlighted in the entire manuscript. Suggest to rephrase the idea here.

Response: Thank you for your careful review here. We have initiated the discussion with a theoretical discussion of the everyday urbanism and how urban studies/science manifests in the remote sensing and spatial big data era (the Introduction section). But we agree with you and we removed the sentence here to avoid further confusion.

Minor changes / typos:

Line 17: Urban scholars are now equipped

Response: modified. Thank you.

Line 22: reviews the development, current status, and future...

Response: modified, thank you.

Figure 1: Font size should be larger.

Response: Thank you for the comment – We enlarge the figure to make the font more readable.

Figure 2: "Big data thinking" - should it be more specific like "use of big data"?

Response: Thank you, we modified this term here.

Lines 135-145: The authors should also mention the concept of "data integration" in this context.

Response: we added a statement that integrating remote sensing and social sensing data in urban science context is also discussed. Thank you.

Line 184: Infrastructural constructions such as...

Response: Modified. Thank you.

Line 190: For "air traffic", do you mean "flights"?

Response: Yes, we modified this. Thank you.

Line 194: Please change the phrase "well the then"?

Response: to shorten this section, this entire sentence is removed. Thank you.

Line 241: "other fields" - what are those fields?

Response: we added the other fields as Land use land cover change detection, water resource management, and forest management.

Line 243-244: Please remove "the then"

Response: Removed. Thank you.

Lines 245-247: It actually depends on the goal and the scale actually, for example, meso-scale and micro-scale applications, all these scales could be feasible.

Response: Thank you. We modified the statement to confine the limited resolution for typical urban applications here.

Line 315: The authors should describe the distinct dimensions of urban forms.

Response: we added more details about the study on Daqing City, China, specifically the study investigates the city proper’s growth with satellite images.

Line 318: critical to support the 

Response: Corrected. Thank you.

Line 328: satellite-based high resolution images

Response: Corrected. Thank you.

Line 472: dataset that

Response: Corrected. Thank you.

Line 556: reasonably inferential

Response: We removed the reasonable here to avoid confusion.

Line 588: the "2.5" of "PM2.5" should be under-scripted.

Response: Corrected. Thank you.

Line 589: plays a significant role...

Response: corrected. Thank you.

Line 594: development, in both socioeconomic and environmental aspects

Response: Corrected. Thank you.

Line 624: Please describe the use and main idea of this "Shannon diversity index"

Response: We have added a brief description and citation to provide a context for the Shannon Diversity Index. Hope this will gain your approval.

Lines 632-633: What are these spatial and spatio-temporal analytical strategies?

Response: We have clarified this ambiguity. They have employed spatial autoregressive model and multiscale geographically weighted regression model. Thank you.

Line 674: immediate violation of assumption

Response: corrected. Thank you.

Line 692: no longer well-defined.

Response: corrected. Thank you.

Line 698: better than the traditional

Response: corrected. Thank you.

Line 699: that were often masked

Response: Corrected. Thank you.

Line 701: urban attractivity, which produced...

Response: corrected. Thank you.

Line 713: remotely sensed image data and...

Response: corrected, thank you.

Line 721: How do you explain / define the word "advanced"?

Response: we removed the word “advanced” to avoid confusion.

Lines 734-735: that everything in world is inter-connected, but...

Response: we changed the sentence to the original citation from Tobler’s work in 1972. We hope the change is acceptable.

Line 743: process at each

Response: corrected. Thank you

Line 758: enables direct calculation and simulation for Bayesian analysis, especially

Response: corrected. Thank you.

Line 762: compared with the reality

Response: corrected. Thank you.

Line 764: remove "an"

Response: we changed the “an” to “a.” Thank you for your careful review.

Lines 762-763: Do the authors mean "data assimilation" here?

Response: we changed the term from “big data narratives” to “spatial big data” to avoid confusion. Thank you for your meticulous comment.

Generally speaking, the manuscript is of high impact, and the ideas and concepts are explained in a very established manner. The authors have done a lot of work before writing this review article. After addressing all the above comments and suggestions / changes, I am sure the review will be much better.

Response: We sincerely thank you for all the insightful and helpful comments that facilitate making the manuscript better. It is with your diligent work that we are able to push through this manuscript to a level of acceptance. Thank you again.

Round 2

Reviewer 1 Report

This manuscript has not been sufficiently improved to warrant publication in Remote Sensing.  This is a review paper,  but  the  review  is  too  simple  in  its current  status.  The authors must compare these methods and analyze them.  Currently, I cannot find any valuable contributions.

Author Response

Response: Thank you for your critical comments. Per your suggestion, we would be greatly appreciated if you could give more specific comments on what “is too simple,” so that we can have specific responses in our revision.

As the review is currently standing, we have gone through thoroughly the Web of Science Database with the designated keywords. We have carefully reviewed the citations we have cited in this review for thorough understanding and commenting prior to writing this review. We spent over a year’s time (from preparing the special issue to finishing writing the review) to prepare the review. We have consulted with many urban scholars, remote sensing scientists, spatial big data practitioners, methodological developers along the way. We understand we all have our own opinions regarding what is “simple” and what is not. It is, however, more conducive to us if specific comments were given so that we can address them.

In addition, we would really appreciate it if you could specify the methods when you commented on “the authors must compare these methods and analyze them.” Please specify what methods and what do you mean by analyze them? As a review article, our job is to provide critical comments on published results and summarize these results to contribute to the remote sensing community for a renewed aspects of urban remote sensing in the new era of remote sensing and spatial big data. To reflect this idea, we have altered our title to be “Urban remote sensing with spatial big data: a literature review and renewed aspect of urban studies in the new era.”

Still, your criticism is well received, and we strive to make changes to hopefully address your concerns. We appreciate your criticism and hope our efforts could gain your understanding and support.

Reviewer 3 Report

The authors have only addressed to some of my comments and suggestions. After careful checking, some comments have not yet been addressed in a proper manner, or the authors might have forgotten to edit in the main text- the points are mentioned again as follows, and the relevant comments were bold:

Original Comment (1): The review as a whole focus on urban areas, however it would be good to include a sub-section near the end of this review, on how big data science and advancement of more potential data sources could benefit the investigation of rural changes and rural development in recent decades.

Problem: The authors have not added a sub-section, or it is still quite unclear in the revised version.

Original Comment (2): In Section 1 (Introduction), the authors should also highlight the relationship and connection between changing land use patterns with city development, the rural-to-urban transition, especially in data limited region / developing cities. Some good references are as follows:

https://www.mdpi.com/2072-4292/13/16/3337

https://www.mdpi.com/2073-445X/11/2/313

https://ecologicalprocesses.springeropen.com/articles/10.1186/s13717-016-0044-6

Problem: The authors have not yet incorporated the highlights and insights of these 3 references into the revised manuscript. Please reformulate again as appropriate.

Original Comment (4): Lines 94-95: The authors cited a reference from Koh et al. [127] here, on urban public health, however this topic is rather far away from the focus of current review report, therefore relevant sentences and references should be removed from the current manuscript.

Response: Thank you for your meticulous review. We further reviewed the relevance of Koh’s work with our review, and we agree with your comments. We have removed relevant expressions in our current revision and hope the revision will gain your support.

Problem: However, the reference is still existing in the main text (Lines 96-97) and in reference (Lines 1201-1203). Therefore, the authors have actually not yet finished updating the manuscript based on the comments made. The concerned statement and reference should be removed.

Problems related to Original Point (20): It is well noted that the authors have shortened these two places to provide more concentrated discussion of big data and big data era. However, again, some proper references should be included in this context, which aims at explaining the recent development and connections between temporal trends and development of "big data".

Problem related to Original Comment (23):

It's good to note that the authors have followed the suggestion and provided more information about the future of urban studies/science and how it will rely on the openness of accumulated big data, and the importance of public engagement in interacting with the information.

However, some proper references should be included as well, for example the following three:

https://www.sciencedirect.com/science/article/abs/pii/S221067072100158X

https://www.mdpi.com/2071-1050/14/18/11461

https://bmjopen.bmj.com/content/11/8/e050167

For minor comments, the followings have not been fully addressed.

(a) The font size of Figure 1 has not been enhanced. It is still rather unclear now.

(b) For the concept of "data integration", the authors have added a statement that integrating remote sensing and social sensing data in urban science context is also discussed. However, some proper references should be included in the manuscript.

https://onlinelibrary.wiley.com/doi/abs/10.1002/9781119625865.ch4

https://www.jstor.org/stable/24537825

Again, the manuscript is of very high impact in related disciplines, like remote sensing, urban studies, city development etc. Therefore, I have high expectations on the quality of this manuscript, and the appropriateness of the contents included. I would look forward to seeing the revised version of the manuscript again, once all aforementioned comments have been properly addressed. Thank you very much for trying to make this manuscript as perfect as possible.

Author Response

The authors have only addressed to some of my comments and suggestions. After careful checking, some comments have not yet been addressed in a proper manner, or the authors might have forgotten to edit in the main text- the points are mentioned again as follows, and the relevant comments were bold:

Original Comment (1): The review as a whole focus on urban areas, however it would be good to include a sub-section near the end of this review, on how big data science and advancement of more potential data sources could benefit the investigation of rural changes and rural development in recent decades.

Problem: The authors have not added a sub-section, or it is still quite unclear in the revised version.

Response: Thank you for your meticulous comment – we indeed forgot to provide more details on this particular comment. We have added a section on big data science and advancement could benefit the investigation of rural changes and rural development within the context of urban science. We hope the addition addresses your concern and gain your full support.

Original Comment (2): In Section 1 (Introduction), the authors should also highlight the relationship and connection between changing land use patterns with city development, the rural-to-urban transition, especially in data limited region / developing cities. Some good references are as follows:

https://www.mdpi.com/2072-4292/13/16/3337

https://www.mdpi.com/2073-445X/11/2/313

https://ecologicalprocesses.springeropen.com/articles/10.1186/s13717-016-0044-6

Problem: The authors have not yet incorporated the highlights and insights of these 3 references into the revised manuscript. Please reformulate again as appropriate.

Response: Thank you again for this comment. We have further revised our introduction section and added the relevant citations. We hope the revision will gain your support.

Original Comment (4): Lines 94-95: The authors cited a reference from Koh et al. [127] here, on urban public health, however this topic is rather far away from the focus of current review report, therefore relevant sentences and references should be removed from the current manuscript.

Problem: However, the reference is still existing in the main text (Lines 96-97) and in reference (Lines 1201-1203). Therefore, the authors have actually not yet finished updating the manuscript based on the comments made. The concerned statement and reference should be removed.

Response: Thank you for your meticulous review. We further reviewed the relevance of Koh’s work with our review, and we agree with your comments. We have removed relevant expressions in our current revision and hope the revision will gain your support.

Problems related to Original Point (20): It is well noted that the authors have shortened these two places to provide more concentrated discussion of big data and big data era. However, again, some proper references should be included in this context, which aims at explaining the recent development and connections between temporal trends and development of "big data".

Response: We have added relevant new references in section 3.2.1 that specifically discuss the recent development and connections between temporal trends and development of “big data” in the context of urban science. We hope the addition addresses your concern.

Problem related to Original Comment (23):

It's good to note that the authors have followed the suggestion and provided more information about the future of urban studies/science and how it will rely on the openness of accumulated big data, and the importance of public engagement in interacting with the information.

However, some proper references should be included as well, for example the following three:

https://www.sciencedirect.com/science/article/abs/pii/S221067072100158X

https://www.mdpi.com/2071-1050/14/18/11461

https://bmjopen.bmj.com/content/11/8/e050167

 Response: Thank you very much for pointing this out – we have added the relevant references in the current revision in section 5. This is very helpful.

For minor comments, the followings have not been fully addressed.

(a) The font size of Figure 1 has not been enhanced. It is still rather unclear now.

Response: we have separated the text from the figure and enlarged the font size to be consistent with the main text. We hope this revision will gain your support.

(b) For the concept of "data integration", the authors have added a statement that integrating remote sensing and social sensing data in urban science context is also discussed. However, some proper references should be included in the manuscript.

https://onlinelibrary.wiley.com/doi/abs/10.1002/9781119625865.ch4

https://www.jstor.org/stable/24537825

Response: Thank you very much for the careful review and guidance. We have added the relevant reference in our discussion of data integration. We hope the revision will gain your support.

Again, the manuscript is of very high impact in related disciplines, like remote sensing, urban studies, city development etc. Therefore, I have high expectations on the quality of this manuscript, and the appropriateness of the contents included. I would look forward to seeing the revised version of the manuscript again, once all aforementioned comments have been properly addressed. Thank you very much for trying to make this manuscript as perfect as possible.

Response: We have tried to address all of your concerns in this round of revision. Your insightful and meticulous review is of enormous help for us to improve the current manuscript. We appreciate the efforts you put in this review and sincerely hope our revision will gain your support.